# Empowering Test-Time Adaptation with Complementary Vision-Language Knowledge in Open-World Scenarios

## Abstract

Test-time adaptation in open-world scenarios (OWTTA), which addresses both domain discrepancy and semantic variance, has gained increasing attention for enabling models to adapt dynamically during inference. Existing approaches mainly rely on discriminative models, whose over-specialized knowledge restricts their adaptability in open-world settings. In contrast, vision-language models (VLMs), trained on diverse large-scale data, provide broader and more transferable knowledge, yet their role in OWTTA remains underexplored. In this work, we propose a framework empowered by vision-language models, termed Vision-Language knowledge Boosted Open-world test-time adaptation (VLBO). Specifically, by casting OWTTA into a probabilistic perspective, we first propose agreement-boosted filtering (AF), in which the discriminative model assumes the primary role of filtering out out-of-distribution samples, while the VLM provides a reinforcing signal to refine this process based on its agreement with the discriminative model. We then introduce semantics-boosted adaptation (SA), where VLM-extracted representations serve as semantic guidance to enhance the discriminative model's adaptation to target domains. This unified framework leverages the complementary strengths of vision-language models and discriminative counterparts, enabling robust and effective adaptation in open-world scenarios. Extensive experiments across multiple benchmarks demonstrate the consistent effectiveness of the proposed method.

## 1 Introduction

Deep learning has witnessed rapid advances and demonstrated strong performance in diverse applications. These achievements are typically realized under the assumption that the training and test data follow the same i.i.d. distribution (Gong et al., 2022). In open-world scenarios, however, this assumption is often violated, as *domain discrepancy* (Qu et al., 2024) and *semantic variance* (Zhao & Lee, 2024) coexist, leading to performance degradation and rendering models unreliable in practice. For instance, in medical image analysis, domain discrepancy may arise from differences in imaging equipment, while semantic variance can occur when the testing data include novel or unseen disease categories. This dual challenge implies that models should not only adapt to domain shifts among in-distribution (ID) samples but also mitigate the negative impact introduced by out-of-distribution (OOD) samples. To address this challenge, open-world test-time adaption (OWTTA) (Li et al., 2023) has been proposed as a promising paradigm, enabling models to adapt to the target domain without requiring label data or source statistics.

Conventional OWTTA methods (Gao et al., 2024) often rely on discriminative models, which can learn task-specific representations but remain constrained by their limited adaptability. Recently, vision-language models (VLMs) (Radford et al., 2021), pre-trained on large-scale and diverse data, have been widely recognized for their zero-shot ability. By jointly modeling vision and language, they learn powerful representations that capture rich semantics and establish effective alignment between visual inputs and textual class names. This property inherently mitigates the impact of domain discrepancy and makes VLMs a strong backbone for TTA (Shu et al., 2022; Karmanov et al., 2024), providing richer semantic cues that enhance generalization across domains. Despite

these advantages, their potential in open-world scenarios remains largely underexplored, since most existing studies fail to consider domain discrepancy and semantic variance simultaneously.

In this paper, we focus on the critical yet underexplored challenge of empowering OWTTA with vision–language models. Specifically, we propose Vision-Language knowledge Boosted Open-world test-time adaptation (VLBO) method, which formulates OWTTA as a probabilistic modeling problem. Based on similarity measures, we propose an agreement-boosted filtering (AF) module to improve the effectiveness of data filtering. Building on this, we further leverage the product-of-experts principle to harness the zero-shot capability of vision–language models, thereby facilitating the adaptation of discriminative models through semantic-boosted adaptation (SA). By working in synergy, AF and SA enable vision–language models to act as a powerful backbone for OWTTA, effectively bridging zero-shot priors with the demands of open-world adaptation. The contributions of this paper are summarized as follows:

- We investigate the challenging yet underexplored problem of empowering open-world test-time adaptation with vision–language models. Building on a series of empirical observations, we reveal the potential of vision–language models in this setting and introduce a probabilistic formulation that serves as the foundation of our framework.
- We develop an agreement-boosted filtering (AF) module that performs data filtering based on both discriminative feature–prototype similarity and vision–language image–text similarity, where the latter is incorporated as weighted auxiliary evidence, providing a more reliable strategy to mitigate the negative impact of OOD samples.
- We propose a semantic-boosted adaptation (SA) module that leverages the rich semantic knowledge embedded in vision–language models as an auxiliary signal to reinforce and boost the adaptation of discriminative models, thereby enabling more robust performance under open-world scenarios.

## 2 RELATED WORK

**Test-Time Adaptation.** Test-time adaptation (TTA) (Su et al., 2024a; 2022) has received increasing attention as it enables models to adapt to target domains using only the source model and unlabeled target data. A variety of strategies have been explored for test-time adaptation, including entropy minimization (Liang et al., 2020; Bar et al., 2024; Zhang et al., 2025; Wu et al., 2025), distribution alignment (Wang et al., 2024b; Zhang et al., 2024; Su et al., 2024b), and continual adaptation (Wang et al., 2022; Niu et al., 2022; Han et al., 2025), which have shown strong effectiveness in the closed-world setting. These methods, however, typically assume that the source and target domains share the same label space, which restricts their applicability in more realistic scenarios.

**Open-World Test-Time Adaptation.** In open-world scenarios, domain discrepancy and semantic variance coexist, making the adaptation task considerably more challenging. Recent attempts to extend TTA into this setting often formulate the problem as a two-step process of OOD detection and adaptation, typically relying on distribution alignment (Li et al., 2023) or entropy-based criteria (Gao et al., 2024; Gong et al., 2023). While these studies still mainly rely on task-specific representations learned by discriminative models, which limits their robustness and generalization ability under open-world conditions.

**Open-Set Domain adaptation.** Compared with open-world test-time adaptation, open-set domain adaptation (OSDA) (Busto & Gall, 2017; Pham et al., 2025; Choe et al., 2024) assumes access to the entire target batch during adaptation and allows the target domain to contain novel categories absent from the source. Recent advances in open-set source-free domain adaptation (OS-SFDA) (Yu et al., 2025; Liu et al., 2025; Wan et al., 2024) further enable adaptation without accessing source data or source statistics. However, under limited target data accessibility, these techniques may face challenges in fully satisfying the needs of open-world scenarios.

**Pre-trained Vision-Language Models.** Large-scale vision–language models (VLMs), including CLIP (Radford et al., 2021), ALIGN (Jia et al., 2021), and GroupViT (Xu et al., 2022), are trained on massive and diverse image–text pairs through self-supervised contrastive learning (Chen et al., 2020). Benefiting from the broad coverage of their pre-training data, these models exhibit strong

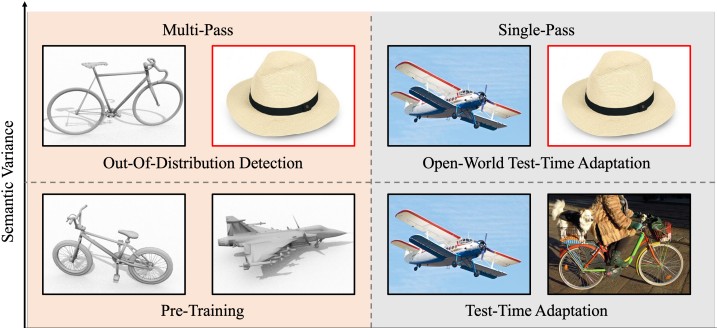

Figure 1: Illustration of the open-world test-time adaptation setting, characterized by the joint presence of *domain discrepancy* and *semantic variance* under a single-pass constraint.

generalization ability across a wide range of downstream tasks. Building on this observation, several works investigate their potential within test-time adaptation settings (Osowiechi et al., 2024; Wang et al., 2024a; Phan et al., 2024). In parallel, some recent studies have also explored leveraging VLMs for OOD detection by finetuning the text encoder (Wang et al., 2023), introducing additional prompts (Jiang et al., 2024), or training a linear classifier (Cao et al., 2025).

## 3 BACKGROUND AND KEY OBSERVATIONS

In this section, we first introduce the protocol of open-world test-time adaptation to clarify the problem formulation and describe the zero-shot classification capability of CLIP. Then, through three empirical observations, we reveal the respective advantages of discriminative models and vision-language models.

### 3.1 BACKGROUND

**Open-World Test-Time Adaptation.** In open-world scenarios, models may encounter not only *domain discrepancy* but also *semantic variance*. This requires them to adapt to domain shifts while mitigating the negative impact of OOD samples, all within a single-pass test-time setting. An illustration of this setting is shown in Fig. 1.

Formally, let $\mathcal{Y}_S = \{1, 2, \ldots, K\}$ denote the label space of known classes, i.e., in-distribution (ID) classes, and $\mathcal{Y}_O$ denote the label space of unknown classes, i.e., out-of-distribution (OOD) classes. Given a discriminative OWTTA model $h_\theta = \{f_\theta, w, b\}$, where $f_\theta$ denotes the encoder and $(w, b)$ parameterize the linear classifier, the goal is to adapt the model to the target distribution while reliably separating ID samples from OOD ones:

$$\begin{aligned} p\left(y, \beta \mid x; h_\theta\right) &= p\left(y, \beta = 1 \mid x; h_\theta\right) + p\left(y, \beta = 0 \mid x; h_\theta\right) \\ &= p\left(\beta = 1 \mid x; h_\theta\right) p\left(y \mid x, \beta = 1; h_\theta\right), \end{aligned} \tag{1}$$

where $y \in \mathcal{Y}_S$ is the ID class label variable and $\beta \in 0, 1$ is an indicator variable distinguishing between ID and OOD samples, i.e., $\beta = 1$ for ID and $\beta = 0$ for OOD. It is trivial to have $p\left(y, \beta = 0 \mid x, ; h_\theta\right) = 0$ since OOD samples cannot be classified into $\mathcal{Y}_S$. The first term $p\left(\beta = 1 \mid x; h_\theta\right)$ corresponds to data filtering and the second term $p\left(y, \beta = 1 \mid x; h_\theta\right)$ models the ID adaptation process.

**Zero-Shot Classification with CLIP.** By leveraging its dual-encoder architecture, CLIP can perform zero-shot classification directly with only class names. Let $F = \{f_I, f_T\}$ denotes CLIP with image encoder $f_I$ and text encoder $f_T$. Given a set of candidate categories, we first construct descriptive text prompts $e_j$ for each class using a collection of natural language templates. These prompts are encoded by the text encoder $f_T$ to obtain the textual representation $f_T(e_j)$. For an input image $x$, its visual representation is extracted with the image encoder $f_I(x)$. Zero-shot classification

is then performed by computing the similarity between the visual and textual representations, i.e.,

$$p(y = j \mid x; F) = \frac{\exp\left(f_T(e_j)^\top f_I(x)/\kappa\right)}{\sum_{i=1}^{K} \exp\left(f_T(e_i)^\top f_I(x)/\kappa\right)}, \tag{2}$$

where $\kappa$ stands for the temperature. Finally, the predicted label is obtained by selecting the class with the maximum probability.

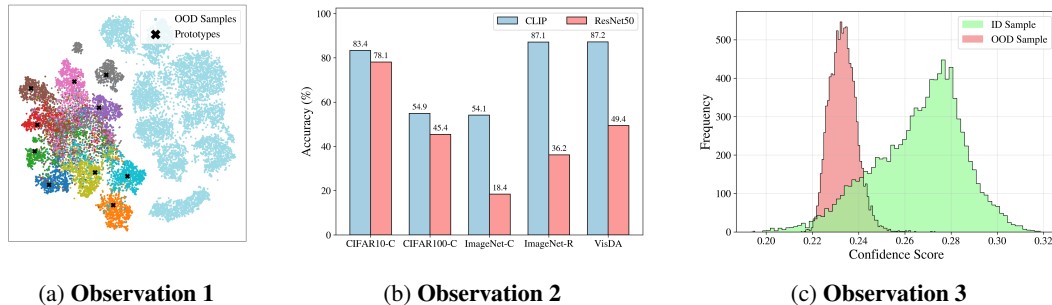

| (a) **Observation 1** | (b) **Observation 2** | (c) **Observation 3** |

Figure 2: (a) **Observation 1** shows the t-SNE visualization of ResNet on the CIFAR10-C & MNIST open-world dataset, revealing that the discriminative model has captured task-specific information. (b) **Observation 2** reports the classification accuracy on ID datasets, where CLIP consistently outperforms ResNet under domain discrepancy, highlighting its superior semantic representations for bridging domain gaps. (c) **Observation 3** is derived from the cosine similarity analysis of CLIP on the CIFAR10-C & MNIST open-world dataset, which reveals its ability to capture semantic deviations.

### 3.2 KEY OBSERVATIONS IN OPEN-WORLD TEST-TIME ADAPTATION

In this section, we present empirical observations that characterize the behavior of discriminative model ResNet (He et al., 2016) and the vision-language model CLIP under the open-world test-time adaptation setting, with the corresponding text prompts provided in Appendix A.1.

- **Observation 1: Discriminative Model Encodes Task-Specific Features.** We empirically observe that discriminative models, such as ResNet pre-trained on the source domain, preserve a well-formed geometric structure in the target domain, even under domain discrepancy and semantic variance. As shown in Fig. 2a, the learned representations of ID samples cluster around the classifier weights, which serve as prototypes of the source classes. In contrast, OOD samples are located far from all prototypes. It indicates that discriminative models retain task-specific features and provide a reliable backbone for open-world adaptation.

- **Observation 2: Vision-Language Model Preserves ID Accuracy under Discrepancy.** CLIP is well known for its strong zero-shot classification ability, relying solely on class names to generalize across diverse tasks. As shown in Fig. 2b, it outperforms the fully supervised ResNet under domain discrepancy, suggesting that large-scale vision–language models offer superior robustness and semantic generalization, enabling strong ID classification even in challenging conditions.

- **Observation 3: Vision-Language Model Captures Semantic Deviations.** Benefiting from pre-training on large-scale image–text pairs, CLIP acquires robust semantic representations that help distinguish ID from OOD samples. As illustrated in Fig. 2c, OOD samples tend to receive lower confidence scores, indicating that CLIP can partially capture semantic deviations and thus contributes to OOD detection.

According to these observations, ResNet can serve as a task-specific backbone for OWTTA, while CLIP exhibits strong generalization ability in both ID classification and OOD detection. This motivates us to leverage the vision–language knowledge of CLIP to empower the adaptation of discriminative models, achieving more effective open-world test-time adaptation.

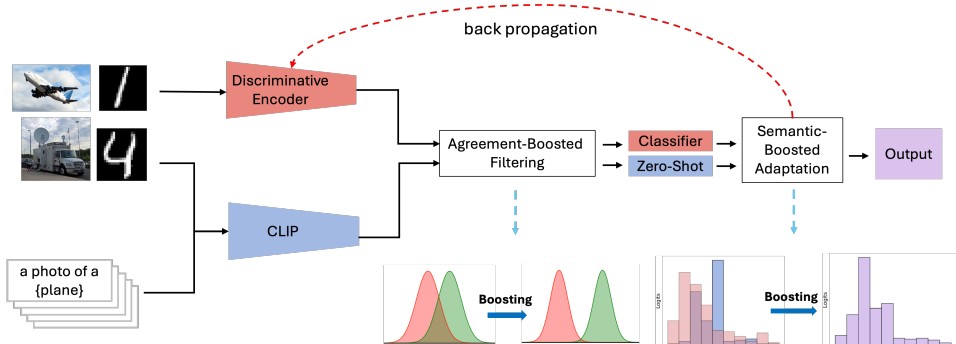

Figure 3: The overall framework of VLBO. Agreement-Boosted Filtering couldshib make OOD samples more distinguishable; Semantic-Boosted Adaptation could corrects false predictions by the VLM and the task-specific discriminative model through their agreement on the input image semantics. See Fig. 5, Fig. 6, Fig. 7, Fig. 8, and Fig. 9 for empirical support.

## 4 METHODOLOGY

In this section, we propose Vision-Language knowledge Boosted Open-world Test-Time Adaptation (VLBO), which is composed of two complementary components: agreement-boosted filtering (AF) and semantic-boosted adaptation (SA). As shown in Fig. 3, VLBO extends the conventional OWTTA framework with discriminative models by incorporating vision–language knowledge from CLIP. Formally, the VLBO objective decomposes into the two complementary components:

$$p\big(y, \beta \mid x; h_\theta, F\big) = \underbrace{p\big(\beta = 1 \mid x; h_\theta, F\big)}_{\text{AF}} \cdot \underbrace{p\big(y \mid x, \beta = 1; h_\theta, F\big)}_{\text{SA}}, \tag{3}$$

where $F$ is the CLIP model. The AF module enhances OOD filtering by integrating task-specific features from ResNet with auxiliary semantic cues from the vision-language model CLIP, enabling more reliable separation of ID and OOD samples. The SA module further adapts ResNet to the target domain using the filtered ID data, leveraging semantic guidance of CLIP to boost classification accuracy and reinforce robustness against domain discrepancy.

### 4.1 AGREEMENT-BOOSTED FILTERING

In the following, we present the agreement-boosted filtering (AF) module, defined by: $p\left(\beta \mid x; h_\theta, F\right)$, where the discriminative model primarily performs OOD filtering, while the vision–language model provides auxiliary semantic signals that reinforces the process through agreement with the discriminative model.

**Confidence Aware Filtering.** We start by estimating $p(\beta \mid x; h_\theta)$ using the discriminative model, which serves as the baseline for filtering OOD samples. According to Observation 1, the ID features generated by the discriminative model tend to be distributed closer to their class prototypes than OOD samples. This property motivates us to model the two types of samples with Gaussian-shaped distributions, providing a principled way to filter out OOD samples.

To model $p(\beta \mid x; h_\theta)$, we first define the confidence score $s$ based on $x$ and $h_\theta$ as the maximum cosine similarity between a feature and the class prototypes:

$$s = \max_j \frac{z^\top c_j}{\|z\|\|c_j\|}, \tag{4}$$

where $z = f_\theta(x)$ is the feature embedding extracted from discriminative model and $c = [c_1, c_2, \ldots, c_K]^\top \in \mathbb{R}^{K \times d}$ denotes the prototypes initialized by the classifier weights $w$.

Based on Observation 1, we utilize the confidence score to distinguish ID and OOD. With its unsupervised manner, we formulate a simple 1D clustering task with two classes. Specifically, $s$ is viewed as drawn from a mixture of two Gaussian distributions with the same variance $\sigma$:

$s \in \pi^+ \mathcal{N}(\mu^+, \sigma^2) + \pi^- \mathcal{N}(\mu^-, \sigma^2)$, where $\pi^+$ and $\pi^-$ are the mixing coefficients, $\mu^+$ is the mean for ID samples and $\mu^-$ is the mean for OOD samples. Consequently,

$$p(\beta = 1 \mid s; \mu^+, \mu^-, \pi^+, \pi^-, \sigma) = \frac{\pi^+ \exp\left(-\frac{1}{2\sigma}(s - \mu^+)^2\right)}{\pi^+ \exp(-\frac{1}{2\sigma}(s - \mu^+)^2) + \pi^- \exp(-\frac{1}{2\sigma}(s - \mu^-)^2)}, \quad (5)$$

$$p(\beta = 0 \mid s; \mu^+, \mu^-, \pi^+, \pi^-, \sigma) = \frac{\pi^+ \exp(-\frac{1}{2\sigma}(s - \mu^-)^2)}{\pi^+ \exp(-\frac{1}{2\sigma}(s - \mu^+)^2) + \pi^- \exp(-\frac{1}{2\sigma}(s - \mu^-)^2)}. \quad (6)$$

Based on the connection between $k$-means and mixtures of Gaussians (Kulis & Jordan, 2012), when $\sigma \to 0$, we can have the following $k$-means based clustering loss:

$$\underset{S^+, S^-}{\arg\min} \; \frac{1}{N^+} \sum_{i \in \mathcal{S}^+} (s_i - \hat{\mu}^+)^2 + \frac{1}{N^-} \sum_{i \in \mathcal{S}^-} (s_i - \hat{\mu}^-)^2, \quad (7)$$

where $i$ is the index of the sample $x_i$ and $s_i$ is its confidence score, $S^+ = \{i \mid \beta_i = 1\}$ is denoted as the ID cluster with $N^+ = |\mathcal{S}^+|$ and $S^- = \{i \mid \beta_i = 0\}$ as the OOD cluster with $N^- = |\mathcal{S}^-|$, $\hat{\mu}^+$ and $\hat{\mu}^-$ are the corresponding means, namely $\hat{\mu}^+ = \frac{1}{N^+} \sum_{i \in \mathcal{S}^+} s_i$, $\hat{\mu}^- = \frac{1}{N^-} \sum_{i \in \mathcal{S}^-} s_i$.

**Agreement-Boosted Filtering.** To improve OOD filtering, we propose the agreement-boosted filtering module $p(\beta \mid x; h_{\theta_t}, F)$, which augments the discriminative model's confidence with semantic cues from CLIP. Following Observation 2, CLIP can capture semantic deviations between ID and OOD samples. We therefore define its confidence as the maximum cosine similarity between image and text features:

$$\hat{s} = \max_j \frac{f_T(e_j)^\top f_I(x)}{\|f_T(e_j)\| \, \|f_I(x)\|}. \quad (8)$$

Since the two confidence scores, $s$ from ResNet and $\hat{s}$ from CLIP, are not directly comparable in scale, we transform them into calibrated evidences to ensure a consistent contribution via a weighted average:

$$\tilde{s} = \frac{s + \alpha \hat{s}}{1 + \alpha}, \quad (9)$$

where $\alpha$ is instantiated as $\alpha = \frac{|\mathcal{S}^+ \cap \mathcal{S}_c^+|}{|\mathcal{S}^+|}$ to quantify the consistency between the discriminative model and CLIP on ID predictions. The boosted confidence score $\tilde{s}$ can then be directly substituted into Eq. (7) to improve the estimation of $\beta$.

## 4.2 SEMANTIC-BOOSTED ADAPTATION

In this subsection, we introduce the semantic-boosted adaptation (SA) module, which leverages vision–language knowledge to enhance the discriminative model's adaptation to the target domain. Formally, SA is modeled as: $p(y \mid x, \beta = 1; h_\theta, F)$, where $F$ provides semantic guidance from vision–language models to boost classification performance and improve robustness against domain discrepancy.

Specifically, for the discriminative model, the likelihood for class $k$ is given by the exponential of the model output:

$$p(y = k \mid x, \beta = 1; h_\theta) \propto \exp\left(w_k^\top f_\theta(x) + b_k\right), \quad (10)$$

where $w_k$ and $b_k$ denote the weight and bias of the classifier for class $k$. For the vision–language model, we define its knowledge of class $k$ as the exponential of the similarity between the corresponding text and image embeddings:

$$p(y = k \mid x, \beta = 1; F) \propto \exp\left(f_T(e_k)^\top f_I(x)\right). \quad (11)$$

Unlike the discriminative model in Eq. (10), restricted to knowledge learned from a specific source domain, the vision-language knowledge originates from large-scale vision–language pre-training. As a result, it encodes broad semantic knowledge and exhibits stronger generalization ability under domain discrepancy, thereby providing complementary guidance to the discriminative model.

In order to boost the adaptation process of the discriminative model, we employ the product-of-experts (PoE) (Hinton, 2002) principle, where the vision-language model provides semantic knowledge to strengthen mutually reinforcing predictions and suppress conflicting ones. Correspondingly, the semantic-boosted adaptation module is defined as:

$$p(y = k \mid x, \beta = 1; h_\theta, F) = \frac{p(y = k \mid x, \beta = 1; h_\theta) \cdot p(y = k \mid x, \beta = 1; F)}{\sum_j p(y = j \mid x, \beta = 1; h_\theta) \cdot p(y = j \mid x, \beta = 1; F)}$$
$$= \sigma\Big(w_k^\top f_\theta(x) + b_k + f_T(e_k)^\top f_I(x)\Big). \tag{12}$$

In this way, the discriminative model contributes task-specific decision boundaries, while the vision–language model empowers the adaptation process with complementary semantic knowledge derived from its strong zero-shot generalization capability, leading to more robust performance under distribution shifts.

Let $t \in \{1, 2, \ldots, T\}$ denote the time index. The overall training objective is defined as:

$$\mathcal{L} = \mathcal{L}_{\text{CE}} + \mathcal{L}_{\text{MSE}} + \lambda \mathcal{L}_{\text{Div}}, \tag{13}$$

where cross-entropy loss drives adaptation with pseudo-labels over $\mathcal{S}^+$:

$$\mathcal{L}_{\text{CE}} = -\frac{1}{N^+} \sum_{i \in \mathcal{S}^+} \log p(y = \hat{y}_i^t \mid x_i^t, \beta = 1; h_{\theta_t}, F), \tag{14}$$

where $\hat{y}_i^t = \arg\max_j \ p(y = j \mid x_i^t, \beta = 1; h_{\theta_t}, F)$ denotes the pseudo-labels of $x_i$, $\forall i \in \mathcal{S}^+$ at time $t$.

The second term encourages feature compactness of the discriminative model with prototypes updated in an exponential moving average (EMA) manner (Pan et al., 2024):

$$\mathcal{L}_{\text{MSE}} = \frac{1}{N^+} \sum_{i \in \mathcal{S}^+} \sum_{j=1}^{K} \xi_{ij}^t \|z_i^t - c_j^t\|_2^2, \tag{15}$$

where $\xi_{ij}^t \in \{0, 1\}$ is a one-hot indicator that assigns each feature $z_i^t$ to its nearest prototype $c_j^t$. In addition, $\mathcal{L}_{\text{Div}} = \sum_{k=1}^{K} \bar{p}^t(k) \log \bar{p}^t(k)$ serves as a regularization term to penalize collapsed predictions, controlled by trade-off parameter $\lambda$, where $\bar{p}(k)^t = \frac{1}{N^+} \sum_{i \in \mathcal{S}^+} p(\hat{y}_i^t = k \mid x_i^t, \beta = 1; h_{\theta_t})$ denotes batch-averaged prediction.

The details of the proposed method are shown in Algorithm 1.

---

**Algorithm 1** VLBO

---

**Require:** Test samples $x$ from the target domain $\mathcal{S}_T$, pre-trained discriminative model $h_\theta$, vision-language model $F$, test prompts $e$.
    **Initialization:** Calculate text embeddings by $f_T(e)$
1: **for** $t \leftarrow 1$ to $T$ **do**
2:     Filter out OOD samples by (7)                ▷ Agreement-boosted filtering (AF)
3:     Update $\theta$ by (13), and update $c$ by EMA     ▷ Semantic-boosted adaptation (SA)
4:     Obtain $\hat{y}^t = \arg\max_j \ p(y = j \mid x^t, \beta = 1; h_{\theta_t}, F)$          ▷ Inference
5: **end for**

---

## 5 EXPERIMENTS

### 5.1 EXPERIMENT SETUP

**Datasets.** We conduct evaluations on blur corruption and style transfer benchmarks. For blur corruption, CIFAR10-C, CIFAR100-C, and ImageNet-C (Hendrycks & Dietterich, 2019) are used as ID datasets, while for style transfer, ImageNet-R (Hendrycks et al., 2021) and VisDA-C (Peng et al., 2017) are adopted as ID datasets. The OOD datasets include Gaussian Noise together with five

real-world datasets that exhibit distributions different from the corresponding ID datasets, including MNIST (Deng, 2012), SVHN (Netzer et al., 2011), Tiny-ImageNet (Le & Yang, 2015), CIFAR100-C, and CIFAR10-C. The open-world datasets are constructed by maintaining an equal number of ID and OOD samples. Further dataset specifications are provided in Appendix A.2.

**Evaluation Settings.** Following recent OWTTA studies (Li et al., 2023), we evaluate the models using three complementary metrics: $ACC_I$, $ACC_O$, and $ACC_H$. Specifically, $ACC_I$ denotes the classification accuracy on ID samples, $ACC_O$ measures the detection accuracy on OOD samples, and $ACC_H$ is their harmonic mean. Formal metric definitions are given in Appendix A.3.

**Baselines.** We introduce two-types of baselines for comparison: discriminative model-based baselines and vision-language model-based baselines. The discriminative model-based baselines adopt ResNet50 as the representative architecture, which includes: TEST, BN (Ioffe & Szegedy, 2015), TENT (Wang et al., 2021), SHOT (Liang et al., 2020), OSTTA (Lee et al., 2023), EATA (Niu et al., 2022), CoTTA (Wang et al., 2022), UniEnt (Gao et al., 2024), OWT3 (Li et al., 2023). The vision–language model baselines are constructed upon CLIP, including CLIP-ViT-B/16 (Radford et al., 2021), C-TPT (Yoon et al., 2024), CLIPN (Wang et al., 2023), NegLabel (Jiang et al., 2024), WATT (Osowiechi et al., 2024). More baseline details can be found in Appendix A.4.

## 5.2 Experimental Results and Analysis

In this section, we conduct extensive experiments on five open-world datasets to validate the effectiveness of the proposed method under the OWTTA setting. The results are summarized in Tables 1,2,3, and 4, with the best performance highlighted in **bold** and the second best underlined. From these tables, we can draw the following key observations:

Table 1: Open-world test-time adaptation results on CIFAR10-C and CIFAR100-C.

| CIFAR10-C | Noise | | | MNIST | | | SVHN | | | Tiny | | | CIFAR100-C | | |
|---|---|---|---|---|---|---|---|---|---|---|---|---|---|---|---|
| | $ACC_I$ | $ACC_O$ | $ACC_H$ | $ACC_I$ | $ACC_O$ | $ACC_H$ | $ACC_I$ | $ACC_O$ | $ACC_H$ | $ACC_I$ | $ACC_O$ | $ACC_H$ | $ACC_I$ | $ACC_O$ | $ACC_H$ |
| TEST | 69.19 | 99.96 | 81.78 | 63.61 | 91.11 | 74.92 | 63.54 | 90.91 | 74.80 | 61.09 | 77.48 | 68.32 | 61.27 | 76.58 | 68.07 |
| BN | 39.44 | 60.57 | 47.77 | 60.19 | 69.56 | 64.54 | 69.21 | 80.17 | 74.29 | 69.02 | 75.15 | 71.95 | 68.83 | 73.33 | 71.01 |
| TENT | 42.75 | 61.99 | 50.60 | 63.21 | 72.80 | 67.67 | 70.84 | 82.92 | 76.41 | 69.70 | 76.23 | 72.82 | 69.22 | 74.07 | 71.56 |
| SHOT | 42.71 | 60.72 | 50.15 | 64.57 | 73.51 | 68.75 | 71.12 | 84.37 | 77.18 | 69.51 | 77.06 | 73.09 | 69.10 | 74.76 | 71.82 |
| OSTTA | 53.43 | 13.69 | 21.80 | 53.88 | 36.94 | 43.83 | 62.76 | 41.20 | 49.74 | **74.85** | 32.38 | 45.20 | **76.39** | 29.80 | 42.87 |
| EATA | 40.65 | 64.14 | 49.76 | 58.04 | 72.63 | 64.52 | 66.62 | 82.29 | 73.63 | 66.72 | 77.37 | 71.65 | 66.83 | 75.00 | 70.68 |
| CoTTA | 35.89 | 68.20 | 47.03 | 60.20 | 69.25 | 64.41 | 69.20 | 80.11 | 74.26 | 69.07 | 75.14 | 71.98 | 68.67 | 73.65 | 71.07 |
| UniEnt | 39.29 | 59.97 | 47.48 | 59.70 | 69.79 | 64.35 | 68.47 | 80.33 | 73.93 | 68.58 | 75.46 | 71.86 | 68.80 | 73.50 | 71.07 |
| OWT3 | 69.41 | 99.96 | 81.93 | 63.89 | 91.10 | 75.11 | 63.72 | 90.97 | 74.94 | 61.05 | 77.56 | 68.32 | 61.37 | 76.47 | 68.09 |
| CLIP-ViT-B/16 | 57.82 | 99.97 | 73.27 | 62.31 | 99.69 | 76.69 | 66.05 | 96.70 | 78.49 | 72.76 | 81.27 | 76.78 | 65.17 | 75.68 | 70.03 |
| +C-TPT | 68.36 | 99.97 | 81.20 | 61.89 | 99.69 | 76.37 | 67.50 | 96.70 | 78.10 | 71.32 | 81.27 | 75.97 | 65.22 | 75.68 | 70.06 |
| +CLIPN | 20.39 | 74.72 | 32.04 | 20.41 | 78.84 | 32.42 | 20.48 | 77.15 | 32.37 | 20.52 | 79.97 | 32.66 | 20.53 | 79.38 | 32.62 |
| +NegSample | 45.52 | 100.00 | 62.56 | 45.52 | 99.86 | 62.53 | 45.52 | 96.08 | 61.77 | 45.52 | 94.47 | 61.44 | 45.52 | 91.29 | 60.75 |
| +WATT | 57.49 | 99.97 | 73.00 | 62.23 | 99.69 | 76.63 | 66.69 | 96.70 | 78.94 | 72.70 | 81.27 | 76.75 | 65.71 | 75.68 | 70.34 |
| +Ours | **85.83** | 100.00 | **92.37** | **86.69** | 97.52 | **91.79** | **83.98** | 95.45 | **89.35** | 71.03 | 84.76 | **77.92** | 71.41 | 78.78 | **74.91** |
| CIFAR100-C | Noise | | | MNIST | | | SVHN | | | Tiny | | | CIFAR10-C | | |
| | $ACC_I$ | $ACC_O$ | $ACC_H$ | $ACC_I$ | $ACC_O$ | $ACC_H$ | $ACC_I$ | $ACC_O$ | $ACC_H$ | $ACC_I$ | $ACC_O$ | $ACC_H$ | $ACC_I$ | $ACC_O$ | $ACC_H$ |
| TEST | 25.30 | 99.99 | 40.38 | 24.76 | 46.81 | 32.39 | 26.34 | 91.65 | 40.92 | 26.20 | 86.68 | 40.24 | 25.79 | 87.86 | 39.88 |
| BN | 21.22 | 93.29 | 34.58 | 26.09 | 93.21 | 40.77 | 30.60 | 95.42 | 46.34 | 31.41 | 89.69 | 46.53 | 31.95 | 89.93 | 47.15 |
| TENT | 24.45 | 95.75 | 38.95 | 27.50 | 91.49 | 42.29 | 32.05 | 95.15 | 47.95 | 31.33 | 89.64 | 46.43 | 32.08 | 89.93 | 47.29 |
| SHOT | 25.98 | 96.62 | 40.95 | 28.26 | 87.36 | 42.71 | 32.42 | 95.14 | 48.36 | 31.18 | 89.71 | 46.28 | 32.07 | 90.10 | 47.30 |
| OSTTA | 12.20 | 40.58 | 18.76 | 17.97 | 46.56 | 25.93 | 33.33 | 62.59 | 43.50 | 36.81 | 52.15 | 43.16 | 42.75 | 43.67 | 43.21 |
| EATA | 22.73 | 94.06 | 36.61 | 25.58 | 92.08 | 40.06 | 29.57 | 96.20 | 45.24 | 30.35 | 90.03 | 45.40 | 31.44 | 89.79 | 46.57 |
| CoTTA | 20.93 | 93.25 | 34.19 | 25.23 | 92.85 | 39.68 | 30.09 | 95.26 | 45.73 | 31.17 | 89.85 | 46.28 | 31.86 | 89.90 | 47.05 |
| UniEnt | 20.05 | 94.48 | 33.08 | 25.54 | 92.86 | 40.06 | 31.36 | 94.48 | 47.09 | 31.20 | 89.19 | 46.23 | 32.52 | 89.03 | 47.64 |
| OWT3 | 25.54 | 99.99 | 40.69 | 24.51 | 45.51 | 31.86 | 26.48 | 91.70 | 41.09 | 26.34 | 86.68 | 40.40 | 25.93 | 87.82 | 40.04 |
| CLIP-ViT-B/16 | 33.67 | 2.99 | 5.49 | 31.14 | 99.29 | 47.41 | 31.57 | 94.88 | 47.38 | 37.81 | 66.30 | 48.16 | 31.13 | 68.01 | 42.71 |
| +C-TPT | 6.21 | 2.99 | 5.62 | 21.33 | 99.29 | 35.12 | 21.63 | 94.88 | 35.23 | 29.86 | 66.30 | 41.18 | 21.80 | 68.01 | 33.02 |
| +CLIPN | 12.55 | 74.95 | 21.51 | 12.79 | 76.43 | 21.91 | 12.80 | 73.87 | 21.82 | 12.77 | 74.48 | 21.80 | 12.80 | 73.63 | 21.81 |
| +NegSample | 28.33 | 96.19 | 43.77 | 28.33 | 92.19 | 43.34 | 28.33 | 51.53 | 36.56 | 28.33 | 79.01 | 41.71 | 28.33 | 64.74 | 39.41 |
| +WATT | 37.07 | 2.99 | 5.53 | 33.25 | 99.29 | 49.82 | 33.27 | 94.88 | 49.27 | **41.38** | 66.30 | 50.96 | 33.21 | 68.01 | 44.63 |
| +Ours | **53.78** | 99.91 | **69.92** | **46.82** | 95.68 | **62.87** | **44.44** | 81.66 | **57.56** | 39.84 | 74.37 | **51.89** | **35.92** | 76.02 | **48.79** |

(1) The proposed VLBO consistently outperforms all baselines, validating its effectiveness for OWTTA. In particular, on the ImageNet-C open-world dataset as shown in Table 2, our method achieves consistently higher $ACC_H$, with a margin of at least 11% over the second-best method, demonstrating its robustness and reliability.

(2) The discriminative model effectively captures task-specific information. In particular, the TEST method employs a pre-trained ResNet-50 for OWTTA. As reported in Table 1, its performance can even surpass certain vision-language model-based baselines. This is attributed to the task-specific knowledge learned in the source domain, which enables the model to maintain a degree of stability under domain discrepancy, consistent with our Observation 1.

Table 2: Open-world test-time adaptation results on ImageNet-C.

| ImageNet-C | Noise | | | MNIST | | | SVHN | | |
|---|---|---|---|---|---|---|---|---|---|
| | $ACC_I$ | $ACC_O$ | $ACC_H$ | $ACC_I$ | $ACC_O$ | $ACC_H$ | $ACC_I$ | $ACC_O$ | $ACC_H$ |
| TEST | 11.79 | 40.85 | 18.30 | 11.33 | 59.57 | 19.04 | 11.62 | 82.04 | 20.36 |
| BN | 9.73 | 66.50 | 16.98 | 15.20 | 74.24 | 25.23 | 16.24 | 77.93 | 26.88 |
| TENT | 9.31 | 68.49 | 16.39 | 15.26 | 72.67 | 25.22 | 17.28 | 78.79 | 28.34 |
| SHOT | 9.70 | 64.57 | 16.87 | 14.76 | 71.69 | 24.48 | 17.70 | 79.63 | 28.96 |
| OSTTA | 13.90 | 65.34 | 22.92 | 20.54 | 57.34 | 30.25 | 21.93 | 59.43 | 32.04 |
| EATA | 15.92 | 88.86 | 27.00 | 20.72 | 93.21 | 33.90 | 20.69 | 93.14 | 33.86 |
| CoTTA | 8.75 | 67.48 | 15.49 | 13.78 | 82.82 | 23.63 | 14.13 | 74.66 | 23.76 |
| UniEnt | 8.29 | 73.44 | 14.90 | 14.13 | 79.71 | 24.00 | 14.81 | 82.88 | 25.13 |
| OWT3 | 9.76 | 68.03 | 17.07 | 15.41 | 75.29 | 25.58 | 16.45 | 78.67 | 27.21 |
| CLIP-ViT-B/16 | 27.88 | 100.00 | 43.61 | 29.94 | 99.96 | 46.08 | 29.75 | 96.39 | 45.47 |
| +C-TPT | 15.62 | 100.00 | 27.02 | 11.85 | 99.96 | 21.19 | 13.22 | 96.39 | 23.25 |
| +CLIPN | 7.76 | 74.91 | 14.07 | 7.76 | 77.07 | 14.11 | 7.76 | 75.24 | 14.08 |
| +NegSample | 34.57 | 99.88 | 51.36 | 34.57 | 60.70 | 44.05 | 34.57 | 21.86 | 26.78 |
| +WATT | 28.73 | 100.00 | 44.64 | 30.99 | 99.96 | 47.31 | 30.80 | 96.39 | 46.68 |
| +Ours | 46.01 | 100.00 | 63.03 | 46.85 | 99.00 | 63.60 | 45.58 | 98.20 | 62.26 |

Table 3: Open-world test-time adaptation results on ImageNet-R.

| ImageNet-R | Noise | | | MNIST | | | SVHN | | |
|---|---|---|---|---|---|---|---|---|---|
| | $ACC_I$ | $ACC_O$ | $ACC_H$ | $ACC_I$ | $ACC_O$ | $ACC_H$ | $ACC_I$ | $ACC_O$ | $ACC_H$ |
| TEST | 24.71 | 100.00 | 39.63 | 23.60 | 99.81 | 38.17 | 22.46 | 98.89 | 36.61 |
| BN | 16.38 | 89.76 | 27.70 | 18.70 | 82.97 | 30.52 | 19.03 | 86.08 | 31.17 |
| TENT | 16.87 | 93.23 | 28.57 | 19.81 | 84.72 | 32.11 | 20.09 | 87.99 | 32.71 |
| SHOT | 17.27 | 96.24 | 29.28 | 19.70 | 81.43 | 31.72 | 20.44 | 89.61 | 33.29 |
| OSTTA | 25.46 | 58.59 | 35.50 | 22.46 | 49.67 | 30.93 | 25.66 | 58.02 | 35.58 |
| EATA | 16.21 | 91.71 | 27.55 | 18.95 | 86.24 | 31.07 | 18.95 | 88.37 | 31.21 |
| CoTTA | 15.58 | 85.41 | 26.35 | 17.40 | 86.71 | 28.98 | 17.51 | 83.04 | 28.92 |
| UniEnt | 10.56 | 72.46 | 18.43 | 15.01 | 79.59 | 25.26 | 15.84 | 83.72 | 26.64 |
| OWT3 | 16.56 | 92.13 | 28.07 | 19.13 | 83.75 | 31.15 | 19.42 | 86.93 | 31.75 |
| CLIP-ViT-B/16 | 46.43 | 100.00 | 63.42 | 55.14 | 99.85 | 71.05 | 60.98 | 99.16 | 75.52 |
| +C-TPT | 43.86 | 100.00 | 60.98 | 52.66 | 99.85 | 68.95 | 59.20 | 99.16 | 74.14 |
| +CLIPN | 16.75 | 92.83 | 28.38 | 19.16 | 87.04 | 31.41 | 19.20 | 85.32 | 31.35 |
| +NegSample | 63.29 | 100.00 | 77.52 | 63.29 | 96.32 | 76.39 | 63.29 | 89.98 | 74.31 |
| +WATT | 45.91 | 100.00 | 62.93 | 54.84 | 99.85 | 70.80 | 60.65 | 99.16 | 75.27 |
| +Ours | 66.48 | 99.04 | 79.56 | 64.30 | 97.63 | 77.53 | 64.83 | 97.76 | 77.96 |

(3) The proposed agreement-boosted filtering module effectively mitigates unreliable outcomes. For instance, as reported in Table 1, CLIP alone exhibits limited reliability in distinguishing Noise OOD samples on the CIFAR100-C open-world dataset. In contrast, our method reduces the influence of disagreements between models, thereby ensuring more stable and reliable adaptation.

(4) The update of vision representations is essential. Existing CLIP-based baselines primarily rely on the image–text alignment property of CLIP. Most of them focus on modifying the text encoder, such as introducing additional prompts, calibrating prompt embeddings, or adding extra branches, while leaving the vision representation unchanged. Consequently, the absence of visual representation update limits their robustness under domain discrepancy. In contrast, our method employs CLIP as auxiliary semantic guidance to enhance the discriminative model, explicitly updating its representation and thereby enabling more reliable adaptation.

In conclusion, our approach introduces a vision–language–empowered paradigm that effectively exploits the complementary strengths of CLIP and discriminative models, yielding consistently superior performance on open-world datasets.

Table 4: Open-world test-time adaptation results on VisDA-C.

| VisDA-C | Noise | | | MNIST | | | SVHN | | |
|---|---|---|---|---|---|---|---|---|---|
| | $ACC_I$ | $ACC_O$ | $ACC_H$ | $ACC_I$ | $ACC_O$ | $ACC_H$ | $ACC_I$ | $ACC_O$ | $ACC_H$ |
| TEST | 41.27 | 100.00 | 58.43 | 43.12 | 97.41 | 59.78 | 42.06 | 99.46 | 59.12 |
| BN | 45.48 | 99.64 | 62.45 | 42.06 | 79.96 | 55.12 | 42.80 | 88.77 | 57.75 |
| TENT | 51.57 | 100.00 | 68.05 | 41.50 | 79.96 | 54.64 | 44.57 | 88.40 | 59.26 |
| SHOT | 42.39 | 99.94 | 59.53 | 42.69 | 68.08 | 52.48 | 43.45 | 72.69 | 54.33 |
| OSTTA | 39.60 | 54.33 | 45.81 | 46.41 | 29.81 | 36.30 | 44.91 | 45.55 | 45.23 |
| EATA | 43.12 | 99.68 | 60.20 | 41.49 | 81.13 | 54.90 | 41.75 | 89.25 | 56.89 |
| CoTTA | 44.01 | 99.56 | 61.04 | 40.71 | 83.91 | 54.82 | 40.89 | 89.86 | 56.20 |
| UniEnt | 54.84 | 95.99 | 69.80 | 41.92 | 80.13 | 55.04 | 42.83 | 88.53 | 57.73 |
| OWT3 | 46.12 | 99.99 | 63.12 | 42.14 | 77.39 | 54.57 | 43.04 | 91.60 | 58.56 |
| CLIP-ViT-B/16 | 48.51 | 100.00 | 65.33 | 59.81 | 99.71 | 74.77 | 60.08 | 97.35 | 74.30 |
| +C-TPT | 65.15 | 100.00 | 78.90 | 60.16 | 99.71 | 75.04 | 60.17 | 97.35 | 74.37 |
| +CLIPN | 21.64 | 72.88 | 33.37 | 21.66 | 72.61 | 33.37 | 21.64 | 77.04 | 33.79 |
| +NegSample | 39.50 | 100.00 | 56.63 | 39.50 | 99.97 | 56.63 | 39.50 | 99.40 | 56.54 |
| +WATT | 48.60 | 100.00 | 65.41 | 59.93 | 99.71 | 74.86 | 60.06 | 97.35 | 74.29 |
| +Ours | 75.09 | 99.63 | 85.64 | 71.44 | 99.49 | 83.16 | 77.88 | 98.70 | 87.06 |

## 5.3 ABLATION STUDY

In this subsection, we conduct comprehensive ablation studies to assess the contributions of agreement-boosted filtering (AF), semantic-boosted adaptation (SA), the MSE loss $\mathcal{L}_{\mathrm{MSE}}$, and the regularization term $\mathcal{L}_{\mathrm{Div}}$. Specifically, experiments are performed across all open-world datasets, with performance evaluated using the $ACC_H$ metric. The results on ImageNet-C and VisDA-C open-world datasets are summarized in Table 5, while the remaining results are deferred to Appendix A.6. From these tables, we draw three main observations: (1) The proposed SA module brings substantial performance gains. For example, as reported in Table 5, $ACC_H$ improves by 26.73%, 55.77%, and 28.6% on Noise, MNIST, and SVHN, respectively. These results highlight that leveraging vision–language knowledge significantly strengthens the adaptation ability of discriminative models in open-world scenarios. (2) The incorporation of the MSE loss encourages features to align more closely with their class prototypes, yielding compact and well-structured feature spaces. This alignment stabilizes adaptation and enhances the robustness of the discriminative model. (3) The complete VLBO framework consistently achieves superior results across all open-world datasets, validating the soundness of our overall design.

## 5.4 COMPUTATIONAL OVERHEAD ANALYSIS

In this subsection, we evaluate the computational overhead of CLIP-based methods. We use a batch size of 128 and measure both the average running time (s) and the average memory consump-

Table 5: Ablation study on ImageNet-C open-world dataset.

| AF | SA | $\mathcal{L}_{\text{MSE}}$ | $\mathcal{L}_{\text{Div}}$ | Noise | MNIST | SVHN |
|----|----|-----|-----|-------|-------|------|
| × | × | × | ✓ | 5.24 | 4.57 | 5.91 |
| × | × | ✓ | ✓ | 35.25 | 6.79 | 33.28 |
| ✓ | × | ✓ | ✓ | 55.34 | 12.83 | 55.20 |
| × | ✓ | ✓ | ✓ | 61.98 | 62.56 | 61.88 |
| ✓ | ✓ | ✓ | ✓ | **63.03** | **63.60** | **62.26** |

Table 6: Ablation study on VisDA-C open-world dataset.

| AF | SA | $\mathcal{L}_{\text{MSE}}$ | $\mathcal{L}_{\text{Div}}$ | Noise | MNIST | SVHN |
|----|----|-----|-----|-------|-------|------|
| × | × | × | ✓ | 14.58 | 51.78 | 51.86 |
| × | × | ✓ | ✓ | 70.20 | 63.63 | 71.00 |
| ✓ | × | ✓ | ✓ | 71.02 | 66.59 | 71.78 |
| × | ✓ | ✓ | ✓ | 83.93 | 82.21 | 86.77 |
| ✓ | ✓ | ✓ | ✓ | **85.64** | **83.16** | **87.05** |

tion (MB) on all the open-world datasets. The results on ImageNet-C are reported in Table 7, while the remaining results are provided in Appendix A.8. As shown in these results, the proposed method achieves a more favorable trade-off between computational overhead and adaptation performance.

Table 7: Computational overhead comparison on ImageNet-C open-world dataset.

| ImageNet-C | Noise | | | MNIST | | | SVHN | | |
|------------|-------|--------|---------|-------|--------|---------|-------|--------|---------|
| | Time | Memory | $ACC_H$ | Time | Memory | $ACC_H$ | Time | Memory | $ACC_H$ |
| CLIP-ViT-B/16 | 0.2031 | 595.91 | 43.61 | 0.1965 | 595.91 | 46.08 | 0.1961 | 595.91 | 45.47 |
| +C-TPT | 1.3012 | 1387.16 | 27.02 | 1.2929 | 1387.16 | 21.19 | 1.0890 | 1387.15 | 23.25 |
| +CLIPN | 0.3053 | 408.46 | 14.07 | 0.3054 | 408.46 | 14.11 | 0.3060 | 408.46 | 14.08 |
| +NegSample | 0.1262 | 598.33 | 51.36 | 0.1246 | 598.33 | 44.05 | 0.1177 | 598.33 | 26.78 |
| +WATT | 17.8515 | 1102.08 | 44.64 | 18.3008 | 1102.08 | 47.31 | 18.7525 | 1102.08 | 46.68 |
| +Ours | 1.0993 | 1349.36 | 63.03 | 0.9591 | 1337.59 | 63.60 | 0.9623 | 1353.19 | 62.26 |

## 5.5 PARAMETER ANALYSIS

In this subsection, we analyze the effect of the trade-off parameter $\lambda$, which controls the weight of the regularization loss $\mathcal{L}_{\text{Div}}$. The $\mathcal{L}_{\text{Div}}$ term encourages prediction diversity and prevents the model from collapsing to trivial solutions. Specifically, we vary $\lambda$ over the range {1e-4, 5e-4, 1e-3, 5e-3, 1e-2, 5e-2, 1e-1, 5e-1} and report the results in Fig. 4. As shown in the figure, VLBO maintains stable performance across all open-world datasets, demonstrating its robustness to the choice of $\lambda$.

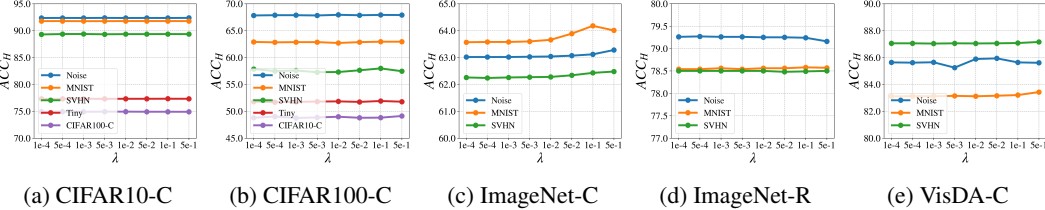

(a) CIFAR10-C  (b) CIFAR100-C  (c) ImageNet-C  (d) ImageNet-R  (e) VisDA-C

Figure 4: Ablation study on the trade-off parameter $\lambda$, showing that VLBO maintains stable performance across all open-world datasets.

## 5.6 CONCLUSION

In this paper, we tackled the challenging problem of open-world test-time adaptation (OWTTA), where both domain discrepancy and semantic variance coexist. We proposed Vision-Language knowledge Boosted OWTTA (VLBO), a unified framework that leverages the complementary strengths of discriminative models and vision–language models (VLMs). VLBO integrates two key components: agreement-boosted filtering (AF), which refines OOD detection by reinforcing discriminative predictions with VLM guidance, and semantic-boosted adaptation (SA), which exploits semantic representations from VLM to improve the adaptation of discriminative models. This paradigm empowers discriminative models with transferable vision–language knowledge, leading to more robust and effective adaptation under open-world conditions. Extensive experiments across diverse benchmarks verify the effectiveness of the proposed VLBO framework.

ETHICS STATEMENT

This work complies with the ICLR Code of Ethics. It involves no potential malicious applications or unintended uses and raises no concerns related to fairness, privacy, security, crowdsourcing, or human subjects.

REPRODUCIBILITY STATEMENT

All datasets used in this paper are publicly available and listed in Appendix A.2. Baseline methods and the modifications required for a fair and consistent comparison are detailed in Appendix A.4. Details on implementation, model architectures, optimization strategies, hyperparameters, and hardware are provided in Appendix A.5, and pseudo-code is included in Algorithm 1 for clarity.

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

# A  APPENDIX

## A.1  LIST OF PROMPTS

The text prompts utilized in our experiments are directly adapted from the template collection introduced in CLIP (Radford et al., 2021):

---

**List of prompts utilized in this paper**

a bad photo of a {class}, a photo of many {class}, a sculpture of a {class}, a photo of the hard to see {class}, a low resolution photo of the {class}, a rendering of a {class}, graffiti of a {class}, a bad photo of the {class}, a cropped photo of the {class}, a tattoo of a {class}, the embroidered {class}, a photo of a hard to see {class}, a bright photo of a {class}, a photo of a clean {class}, a photo of a dirty {class}, a dark photo of the {class}, a drawing of a {class}, a photo of my {class}, the plastic {class}, a photo of the cool {class}, a close-up photo of a {class}, a black and white photo of the {class}, a painting of the {class}, a painting of a {class}, a pixelated photo of the {class}, a sculpture of the {class}, a bright photo of the {class}, a cropped photo of a {class}, a plastic {class}, a photo of the dirty {class}, a jpeg corrupted photo of a {class}, a blurry photo of the {class}, a photo of the {class}, a good photo of the {class}, a rendering of the {class}, a {class} in a video game, a photo of one {class}, a doodle of a {class}, a close-up photo of the {class}, a photo of a {class}, the origami {class}, the {class} in a video game, a sketch of a {class}, a doodle of the {class}, a origami {class}, a low resolution photo of a {class}, the toy {class}, a rendition of the {class}, a photo of the clean {class}, a photo of a large {class}, a rendition of a {class}, a photo of a nice {class}, a photo of a weird {class}, a blurry photo of a {class}, a cartoon {class}, art of a {class}, a sketch of the {class}, a embroidered {class}, a pixelated photo of a {class}, itap of the {class}, a jpeg corrupted photo of the {class}, a good photo of a {class}, a plushie {class}, a photo of the nice {class}, a photo of the small {class}, a photo of the weird {class}, the cartoon {class}, art of the {class}, a drawing of the {class}, a photo of the large {class}, a black and white photo of a {class}, the plushie {class}, a dark photo of a {class}, itap of a {class}, graffiti of the {class}, a toy {class}, itap of my {class}, a photo of a cool {class}, a photo of a small {class}, a tattoo of the {class},

---

## A.2  DATASETS

Our experimental setup follows OWT3 (Li et al., 2023), using the same set of datasets. The details of the ID datasets, including test-set size, number of classes, and corresponding target domains, are summarized in Table 8:

Table 8: In-distribution Datasets Information.

| Datasets | #Images | #Classes | Target Domain |
|----------|---------|----------|---------------|
| CIFAR10-C | 10,000 | 10 | Corruption |
| CIFAR100-C | 10,000 | 100 | Corruption |
| ImageNet-C | 50,000 | 1,000 | Corruption |
| ImageNet-R | 30,000 | 200 | Style Transfer |
| VisDA-C | 55,388 | 12 | Style Transfer |

## A.3  EVALUATION METRIC

Since the OWTTA task consists of both domain discrepancy and semantic variance, we adopt three complementary evaluation metrics, $ACC_I$, $ACC_O$, and $ACC_H$, to provide a more precise assess-

ment of performance. Their definitions are as follows:

$$ACC_I = \frac{\sum_{\{i|y_i \in \mathcal{Y}_S\}} \mathbf{1}\,(\hat{y}_i = y_i)}{|\{i|y_i \in \mathcal{Y}_S\}|},$$

$$ACC_O = \frac{\sum_{\{i|y_i \in \mathcal{Y}_O\}} \mathbf{1}(\hat{y}_i \in \mathcal{Y}_O)}{|\{i|y_i \in \mathcal{Y}_O\}|},$$

$$ACC_H = 2 \cdot \frac{ACC_I \cdot ACC_O}{ACC_I + ACC_O}.$$

Here, $\mathcal{Y}_S$ denotes the set of in-distribution (ID) classes and $\mathcal{Y}_O$ the out-of-distribution (OOD) class. $ACC_I$ evaluates classification accuracy on ID samples, $ACC_O$ measures OOD detection accuracy, and $ACC_H$ provides a balanced evaluation through their harmonic mean.

### A.4   DETAILS OF BASELINES

In the main text, we compare our approach against nine representative baselines built upon discriminative models and four recent methods based on vision–language models. Their details are summarized as follows:

- **TEST**: Direct inference with the pre-trained discriminative model, without any adaptation to the target distribution. This baseline reflects the model's raw generalization ability under OWTTA.

- **BN** (Ioffe & Szegedy, 2015): Batch normalization statistics are updated online during inference, while the rest of the network parameters remain frozen. This allows the model to partially adjust to distribution shifts.

- **TENT** (Wang et al., 2021): The affine parameters (scale and bias) of batch normalization layers are optimized by minimizing the entropy of predictions on test samples, enabling lightweight adaptation through confident predictions.

- **SHOT** (Liang et al., 2020): SHOT fix its classifier during the adaptation, while the feature encoder, pre-trained by source domain data, is updated with self-supervised objectives. It promotes transferable and discriminative representations on target data.

- **OSTTA** (Lee et al., 2023): Low-confidence predictions are filtered as potential OOD samples, and adaptation is performed by entropy minimization on confident ID data, which reduces the risk of error propagation.

- **EATA** (Niu et al., 2022): High-confidence target samples are chosen for adaptation, while Fisher information regularization constrains drastic parameter changes. Together, these mechanisms balance adaptation plasticity and stability.

- **CoTTA** (Wang et al., 2022): Pseudo-labels are averaged across augmentations and weight trajectories to stabilize adaptation, and stochastic restoration of source weights periodically anchors the model to its pre-trained state.

- **UniEnt** (Gao et al., 2024): A Gaussian mixture model is applied to feature–prototype similarities to separate ID and OOD samples. Adaptation is guided by minimizing entropy for ID predictions and maximizing entropy for OOD predictions.

- **OWT3** (Li et al., 2023): Originally designed to align target distributions with source statistics and construct new OOD prototypes. In the OWTTA setting, it is adapted by replacing prototypes with normalized classifier weights and removing the source-alignment loss.

For vision-language model-based baselines, we have:

- **CLIP** (Radford et al., 2021): A vision–language model trained on large-scale image–text pairs with contrastive learning. It performs zero-shot classification by matching image features with text embeddings of class names, providing a strong foundation for open-world adaptation.

- **C-TPT** (Yoon et al., 2024): Builds upon CLIP by refining textual prompts at test time. It optimizes context tokens to better align with target-domain distributions, thereby improving zero-shot generalization under distribution shifts.

- **CLIPN** (Wang et al., 2023): Extends CLIP by introducing an additional text encoder trained with noise-injected supervision on large-scale data. This design explicitly enhances CLIP's capacity for OOD detection, enabling it to better separate in-distribution and out-of-distribution samples.

- **NegSample** (Jiang et al., 2024): Augments CLIP with negative sampling in the joint feature space, where mismatched image–text pairs are explicitly penalized. By enforcing stronger contrast between matched and mismatched pairs, it sharpens the decision boundary between ID and OOD data, thereby enhancing OOD detection capability.

- **WATT** (Osowiechi et al., 2024): Performs test-time adaptation for CLIP by leveraging multiple prompt templates to generate diverse text embeddings. For each template, CLIP produces pseudo labels that are used to update the model. The resulting parameters from different templates are then consolidated through weight averaging, which stabilizes the adaptation process.

In our experiments, some discriminative model baseline methods are not originally tailored for the OWTTA setting. To ensure fair comparison, we follow the protocol in OWT3 (Li et al., 2023). Specifically, we employ ResNet-50 and use Eq. (7) to distinguish ID samples. The compared methods are then applied to the filtered ID subset for subsequent adaptation.

CLIP-based adaptation methods, which were originally developed for the TTA setting, are extended to OWTTA by first detecting OOD samples using CLIP-derived confidence scores and then applying adaptation only to the identified ID subset. CLIP-based OOD detection methods are evaluated in their original form. We report their OOD detection performance directly and compute the ID classification accuracy using either the corresponding trained encoder or the zero-shot CLIP model with updated prompts, depending on the design of each method.

All baseline methods are implemented following the settings and recommended hyperparameters provided in their original papers to ensure a fair and consistent comparison.

## A.5 Implementation Details

All experiments are conducted on a workstation with Ubuntu 22.04.5, Python 3.7.16, and PyTorch 1.13 (Paszke et al., 2019), equipped with an NVIDIA RTX 6000 Ada Generation GPU (48GB memory) and an AMD Ryzen Threadripper PRO 5965WX 24-core CPU. In the proposed VLBO framework, ResNet-50 serves as the backbone and SGD is used as the optimizer. We use SGD optiomizer. For CIFAR10-C and CIFAR100-C open-world datasets, we use a learning rate of $1e-3$ with $\lambda = 0.1$. For ImageNet-C, ImageNet-R, and VisDA-C open-world datasets, the learning rate is reduced to $1e-4$ with $\lambda = 1e-3$.

## A.6 Ablation Study

The main text reports ablation results on the ImageNet-C and VisDA-C open-world datasets. For completeness, we additionally provide the ablation results across all open-world datasets in this section.

Table 9: Ablation study on CIFAR10-C open-world dataset.

| AF | SA | $\mathcal{L}_{\text{MSE}}$ | $\mathcal{L}_{\text{Div}}$ | Noise | MNIST | SVHN | Tiny | CIFAR100-C |
|---|---|---|---|---|---|---|---|---|
| × | × | × | ✓ | 86.09 | 78.47 | 82.48 | 74.00 | 72.71 |
| × | × | ✓ | ✓ | 90.34 | 89.35 | 80.51 | 70.56 | 72.71 |
| ✓ | × | ✓ | ✓ | 90.33 | 89.73 | 87.71 | 76.51 | 74.51 |
| × | ✓ | ✓ | ✓ | 92.33 | 91.44 | 84.48 | 71.00 | 72.95 |
| ✓ | ✓ | ✓ | ✓ | 92.37 | 91.79 | 89.35 | 77.92 | 74.91 |

Table 10: Ablation study on CIFAR100-C open-world dataset.

| AF | SA | $\mathcal{L}_{\text{MSE}}$ | $\mathcal{L}_{\text{Div}}$ | Noise | MNIST | SVHN | Tiny | CIFAR10-C |
|---|---|---|---|---|---|---|---|---|
| × | × | × | ✓ | 49.11 | 43.39 | 46.39 | 46.13 | 46.79 |
| × | × | ✓ | ✓ | 61.43 | 52.47 | 51.35 | 47.28 | 45.78 |
| ✓ | × | ✓ | ✓ | 62.15 | 55.05 | 53.82 | 49.44 | 48.09 |
| × | ✓ | ✓ | ✓ | 68.04 | 61.21 | 50.47 | 48.93 | 48.24 |
| ✓ | ✓ | ✓ | ✓ | 69.92 | 62.87 | 57.56 | 51.89 | 48.79 |

Table 11: Ablation study on ImageNet-R open-world dataset.

| AF | SA | $\mathcal{L}_{\text{MSE}}$ | $\mathcal{L}_{\text{Div}}$ | Noise | MNIST | SVHN |
|----|----|------|------|-------|-------|------|
| × | × | × | ✓ | 44.41 | 40.39 | 37.02 |
| × | × | ✓ | ✓ | 55.37 | 33.17 | 56.95 |
| ✓ | × | ✓ | ✓ | 55.99 | 44.12 | 57.05 |
| × | ✓ | ✓ | ✓ | 78.66 | 75.04 | 76.89 |
| ✓ | ✓ | ✓ | ✓ | 79.56 | 77.53 | 77.96 |

## A.7 BOOST WITH DIFFERENT BACKBONES

In this subsection, we evaluate the proposed method with different backbone architectures to verify the general effectiveness of the VLBO paradigm. Results are reported in Table 12, Table 13, Table 14, Table 15, and Table 16. From these tables, we observe that VLBO consistently outperforms existing methods when using CLIP with a ResNet-50 backbone. Moreover, as shown in Table 12 and Table 13, the vanilla CLIP method fails to reliably filter OOD samples, leading to degraded performance, whereas our method remains effective. These results demonstrate the robustness and effectiveness of VLBO across different architectures.

Table 12: OWTTA results on CIFAR10-C with CLIP ResNet-50 backbone.

| CIFAR10-C | Noise | | | MNIST | | | SVHN | | | Tiny | | | CIFAR100-C | | |
|---|---|---|---|---|---|---|---|---|---|---|---|---|---|---|---|
| | $ACC_I$ | $ACC_O$ | $ACC_H$ | $ACC_I$ | $ACC_O$ | $ACC_H$ | $ACC_I$ | $ACC_O$ | $ACC_H$ | $ACC_I$ | $ACC_O$ | $ACC_H$ | $ACC_I$ | $ACC_O$ | $ACC_H$ |
| CLIP-RN50 | 32.36 | 0.04 | 0.08 | 29.96 | 72.49 | 42.40 | 32.66 | 84.68 | 47.14 | 50.26 | 68.88 | 58.11 | 33.54 | 76.47 | 46.63 |
| +C-TPT | 31.34 | 0.04 | 0.08 | 29.99 | 72.49 | 42.43 | 31.44 | 84.68 | 45.85 | 48.24 | 68.88 | 56.74 | 31.76 | 76.47 | 44.88 |
| +NegSample | 26.01 | 99.99 | 41.28 | 26.01 | 77.95 | 39.00 | 26.01 | 75.86 | 38.74 | 26.01 | 92.66 | 40.62 | 26.01 | 89.75 | 40.33 |
| +Ours | 82.41 | 100.00 | 90.36 | 82.42 | 96.67 | 88.98 | 79.89 | 90.87 | 85.03 | 67.70 | 82.32 | 74.30 | 69.80 | 77.19 | 73.31 |

Table 13: OWTTA results on CIFAR100-C with CLIP ResNet-50 backbone.

| CIFAR100-C | Noise | | | MNIST | | | SVHN | | | Tiny | | | CIFAR10-C | | |
|---|---|---|---|---|---|---|---|---|---|---|---|---|---|---|---|
| | $ACC_I$ | $ACC_O$ | $ACC_H$ | $ACC_I$ | $ACC_O$ | $ACC_H$ | $ACC_I$ | $ACC_O$ | $ACC_H$ | $ACC_I$ | $ACC_O$ | $ACC_H$ | $ACC_I$ | $ACC_O$ | $ACC_H$ |
| CLIP-RN50 | 5.26 | 0.00 | 0.00 | 12.56 | 58.80 | 20.70 | 13.64 | 62.70 | 22.41 | 18.72 | 58.89 | 28.41 | 13.87 | 57.48 | 22.35 |
| +C-TPT | 19.80 | 0.00 | 0.00 | 8.70 | 58.80 | 15.16 | 8.81 | 62.70 | 15.45 | 18.30 | 58.89 | 27.92 | 9.75 | 57.48 | 16.67 |
| +NegSample | 10.68 | 0.00 | 0.00 | 10.68 | 21.56 | 14.28 | 10.68 | 24.17 | 14.81 | 10.68 | 80.58 | 18.86 | 10.68 | 67.45 | 18.44 |
| +Ours | 46.65 | 99.94 | 63.61 | 35.91 | 91.96 | 51.65 | 39.36 | 81.43 | 53.07 | 37.31 | 74.32 | 49.68 | 34.39 | 74.25 | 47.01 |

Table 14: OWTTA results on ImageNet-C with CLIP ResNet-50 backbone.

| ImageNet-C | Noise | | | MNIST | | | SVHN | | |
|---|---|---|---|---|---|---|---|---|---|
| | $ACC_I$ | $ACC_O$ | $ACC_H$ | $ACC_I$ | $ACC_O$ | $ACC_H$ | $ACC_I$ | $ACC_O$ | $ACC_H$ |
| CLIP-RN50 | 9.26 | 100.00 | 16.95 | 12.87 | 99.70 | 22.80 | 13.75 | 98.20 | 24.13 |
| +C-TPT | 7.38 | 100.00 | 13.75 | 14.79 | 99.70 | 25.76 | 17.07 | 98.20 | 29.08 |
| +NegSample | 14.94 | 3.52 | 5.70 | 14.94 | 73.91 | 24.86 | 14.94 | 44.10 | 22.32 |
| +Ours | 39.83 | 99.57 | 56.9 | 38.01 | 98.99 | 54.93 | 39.36 | 98.31 | 56.22 |

Table 15: OWTTA results on ImageNet-R with CLIP ResNet-50 backbone.

| ImageNet-R | Noise | | | MNIST | | | SVHN | | |
|---|---|---|---|---|---|---|---|---|---|
| | $ACC_I$ | $ACC_O$ | $ACC_H$ | $ACC_I$ | $ACC_O$ | $ACC_H$ | $ACC_I$ | $ACC_O$ | $ACC_H$ |
| CLIP-RN50 | 41.12 | 100.00 | 58.28 | 43.11 | 99.94 | 60.24 | 46.42 | 99.49 | 63.31 |
| +C-TPT | 33.62 | 100.00 | 50.32 | 25.51 | 99.66 | 40.62 | 19.17 | 76.21 | 30.63 |
| +NegSample | 44.12 | 100.00 | 61.23 | 44.12 | 99.94 | 61.22 | 44.12 | 97.41 | 60.73 |
| +Ours | 46.02 | 98.71 | 62.78 | 45.30 | 98.64 | 62.09 | 47.30 | 99.56 | 64.13 |

Table 16: OWTTA results on VisDA-C with CLIP ResNet-50 backbone.

| VisDA-C | Noise | | | MNIST | | | SVHN | | |
|---|---|---|---|---|---|---|---|---|---|
| | $ACC_I$ | $ACC_O$ | $ACC_H$ | $ACC_I$ | $ACC_O$ | $ACC_H$ | $ACC_I$ | $ACC_O$ | $ACC_H$ |
| CLIP-RN50 | 39.77 | 99.97 | 56.90 | 48.67 | 98.22 | 65.09 | 50.60 | 94.04 | 65.80 |
| +C-TPT | 37.64 | 99.97 | 54.69 | 34.81 | 64.14 | 45.13 | 32.21 | 92.37 | 47.76 |
| +NegSample | 31.23 | 100.00 | 47.60 | 31.23 | 99.99 | 47.59 | 31.23 | 99.38 | 47.53 |
| +Ours | 60.59 | 99.63 | 75.35 | 55.45 | 99.12 | 71.12 | 62.73 | 98.36 | 76.61 |

## A.8 COMPUTATIONAL OVERHEAD ANALYSIS

The main text reports the computational overhead analysis on the ImageNet-C open-world dataset. For completeness, we further provide the corresponding comparisons on CIFAR10-C, CIFAR100-C, ImageNet-R, and VisDA.

Table 17: Computational overhead comparison on CIFAR10-C open-world dataset.

| CIFAR10-C | Noise | | | MNIST | | | SVHN | | | Tiny | | | CIFAR100-C | | |
|---|---|---|---|---|---|---|---|---|---|---|---|---|---|---|---|
| | Time | Memory | $ACC_H$ | Time | Memory | $ACC_H$ | Time | Memory | $ACC_H$ | Time | Memory | $ACC_H$ | Time | Memory | $ACC_H$ |
| CLIP-ViT-B/16 | 0.1549 | 507.26 | 73.27 | 0.1611 | 507.26 | 76.69 | 0.1573 | 507.26 | 78.49 | 0.1530 | 507.26 | 76.78 | 0.1593 | 507.26 | 70.03 |
| +C-TPT | 0.4697 | 1220.96 | 81.20 | 0.4789 | 1220.96 | 76.37 | 0.4843 | 1220.96 | 78.10 | 0.4948 | 1220.79 | 75.97 | 0.4899 | 1220.79 | 70.06 |
| +CLIPN | 0.3066 | 403.25 | 32.04 | 0.3184 | 403.25 | 32.42 | 0.3171 | 403.25 | 32.37 | 0.3197 | 403.25 | 32.66 | 0.3187 | 403.25 | 32.62 |
| +NegSample | 0.1297 | 578.93 | 62.56 | 0.1130 | 578.93 | 62.53 | 0.1152 | 578.93 | 61.77 | 0.1153 | 578.93 | 61.44 | 0.1137 | 578.93 | 60.75 |
| +WATT | 7.8918 | 1099.38 | 73.00 | 8.8878 | 1099.38 | 76.63 | 10.0848 | 1099.38 | 78.94 | 13.8435 | 1099.38 | 76.75 | 13.0855 | 1099.38 | 70.34 |
| +Ours | 1.0102 | 877.44 | 92.37 | 0.6569 | 782.32 | 91.79 | 1.0926 | 880.54 | 89.35 | 1.2972 | 940.21 | 77.29 | 0.9442 | 878.05 | 74.91 |

Table 18: Computational overhead comparison on CIFAR100-C open-world dataset.

| CIFAR100-C | Noise | | | MNIST | | | SVHN | | | Tiny | | | CIFAR10-C | | |
|---|---|---|---|---|---|---|---|---|---|---|---|---|---|---|---|
| | Time | Memory | $ACC_H$ | Time | Memory | $ACC_H$ | Time | Memory | $ACC_H$ | Time | Memory | $ACC_H$ | Time | Memory | $ACC_H$ |
| CLIP-ViT-B/16 | 0.1555 | 508.36 | 5.49 | 0.1569 | 508.36 | 47.41 | 0.1573 | 508.36 | 47.38 | 0.1556 | 508.36 | 48.16 | 0.1588 | 508.36 | 42.71 |
| +C-TPT | 0.5701 | 1235.24 | 5.62 | 0.4925 | 1235.68 | 35.12 | 0.5178 | 1235.68 | 35.23 | 0.5527 | 1235.53 | 41.18 | 0.5387 | 1235.66 | 33.02 |
| +CLIPN | 0.3161 | 403.69 | 21.51 | 0.3154 | 403.69 | 21.91 | 0.3188 | 403.69 | 21.82 | 0.3186 | 403.69 | 21.80 | 0.3191 | 403.69 | 21.81 |
| +NegSample | 0.1185 | 579.33 | 43.77 | 0.1147 | 579.33 | 43.34 | 0.1142 | 579.33 | 36.56 | 0.1168 | 579.33 | 41.71 | 0.1114 | 579.33 | 39.41 |
| +WATT | 21.5565 | 1099.60 | 5.53 | 7.0037 | 1099.60 | 49.82 | 7.4444 | 1099.60 | 49.27 | 13.5017 | 1099.60 | 50.96 | 10.9796 | 1099.60 | 44.63 |
| +Ours | 0.6530 | 863.94 | 69.92 | 0.5425 | 912.22 | 62.87 | 0.5227 | 774.92 | 57.56 | 0.5583 | 824.13 | 51.89 | 0.5483 | 813.94 | 48.79 |

Table 19: Computational overhead comparison ImageNet-R open-world dataset.

| ImageNet-R | Noise | | | MNIST | | | SVHN | | |
|---|---|---|---|---|---|---|---|---|---|
| | Time | Memory | $ACC_H$ | Time | Memory | $ACC_H$ | Time | Memory | $ACC_H$ |
| CLIP-ViT-B/16 | 0.2026 | 593.34 | 63.42 | 0.1877 | 593.34 | 71.05 | 0.1924 | 593.34 | 75.52 |
| +C-TPT | 0.5452 | 1260.39 | 60.98 | 0.5438 | 1260.39 | 68.95 | 0.5496 | 1260.39 | 74.14 |
| +CLIPN | 0.3019 | 404.49 | 28.38 | 0.3079 | 404.49 | 31.41 | 0.3087 | 404.49 | 31.35 |
| +NegSample | 0.1025 | 596.75 | 77.52 | 0.1195 | 596.75 | 76.39 | 0.1160 | 596.75 | 74.31 |
| +WATT | 7.5363 | 1100.16 | 62.93 | 9.0740 | 1100.16 | 70.80 | 10.3173 | 1100.16 | 75.27 |
| +Ours | 1.2607 | 1156.20 | 79.56 | 0.9742 | 1172.48 | 77.53 | 0.9894 | 1175.29 | 77.96 |

Table 20: Computational overhead comparison on VisDA open-world dataset.

| VisDA | Noise | | | MNIST | | | SVHN | | |
|---|---|---|---|---|---|---|---|---|---|
| | Time | Memory | $ACC_H$ | Time | Memory | $ACC_H$ | Time | Memory | $ACC_H$ |
| CLIP-ViT-B/16 | 0.1829 | 580.69 | 65.33 | 0.1913 | 580.69 | 74.77 | 0.1964 | 580.69 | 74.30 |
| +C-TPT | 0.4648 | 1221.33 | 78.90 | 0.4686 | 1221.33 | 75.04 | 0.4734 | 1221.33 | 74.37 |
| +CLIPN | 0.3049 | 403.56 | 33.37 | 0.3041 | 403.56 | 33.37 | 0.3043 | 403.56 | 33.79 |
| +NegSample | 0.1197 | 596.27 | 56.63 | 0.1143 | 596.27 | 56.63 | 0.1253 | 596.27 | 56.54 |
| +WATT | 6.7990 | 1099.68 | 65.41 | 8.8650 | 1099.68 | 74.86 | 8.1327 | 1099.68 | 74.29 |
| +Ours | 0.6973 | 780.56 | 85.64 | 0.6106 | 770.25 | 83.16 | 0.6254 | 772.57 | 87.06 |

## A.9 VISUALIZATIONS

In this subsection, we present the visualization results of the proposed VLBO method. To intuitively demonstrate the effect of AF, we plot the confidence scores before and after boosting. The corresponding results are shown in Fig. 5, Fig. 6, Fig. 7, Fig. 8, and Fig. 9. As ImageNet-C comprises 1000 classes, we plot a subset of 50 classes to ensure clearer visualization. These figures indicate that AF substantially enlarges the margin between ID and OOD samples, thereby improving their separability. In addition, heatmap visualizations of logits from ResNet and VLBO demonstrate that SA effectively suppresses irrelevant classes while emphasizing the correct class. Overall, these visualizations confirm that AF and SA collaboratively enhance the reliability of adaptation under the OWTTA setting.

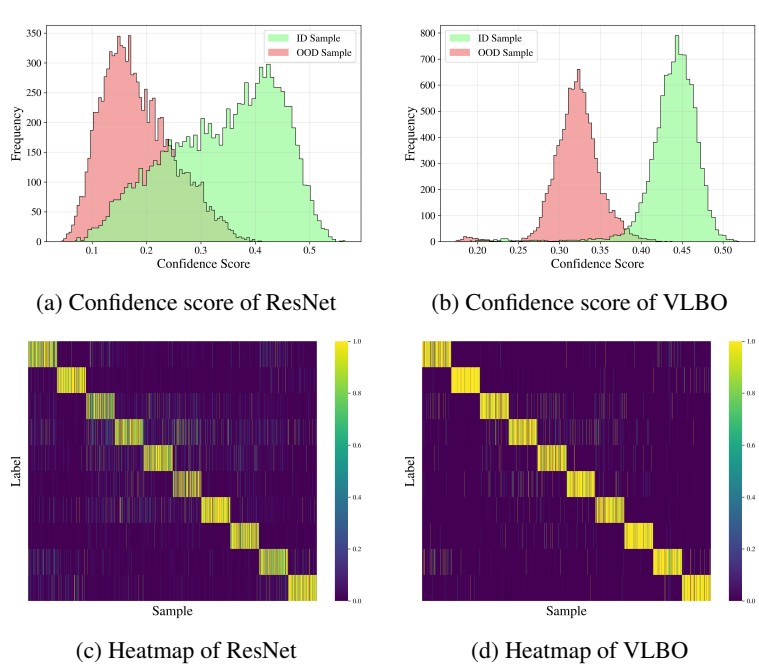

Figure 5: Visualization of OWTTA results on CIFAR10-C & MNIST open-world datasets.

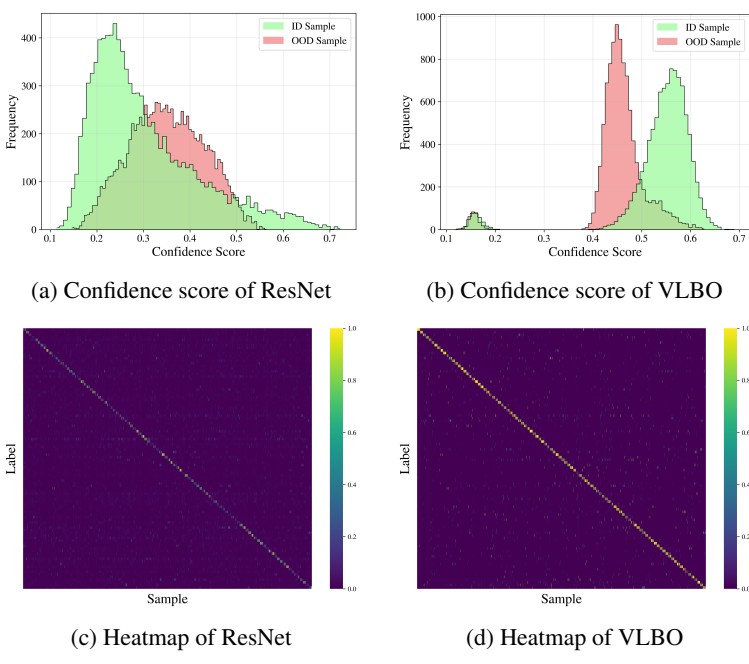

Figure 6: Visualization of OWTTA results on CIFAR100-C & MNIST open-world datasets.

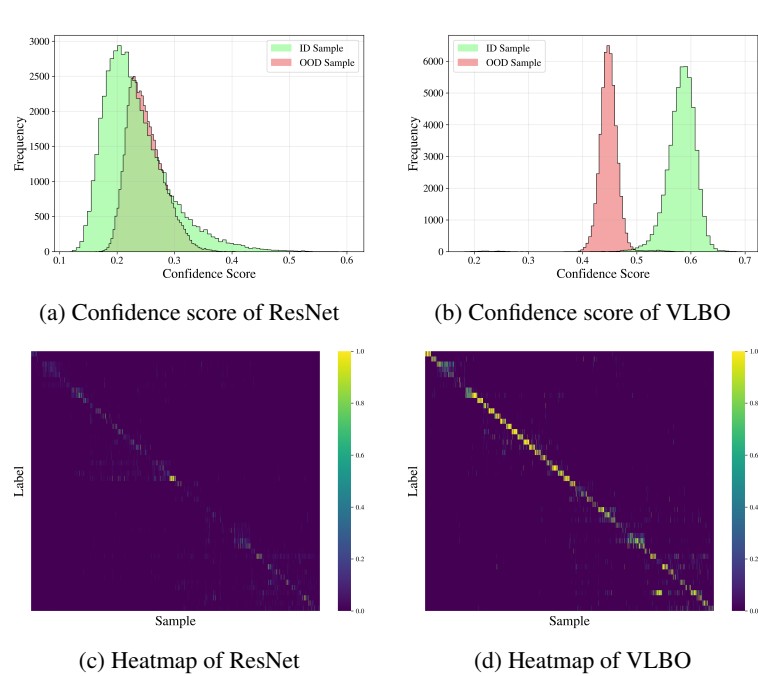

Figure 7: Visualization of OWTTA results on ImageNet-C & MNIST open-world datasets.

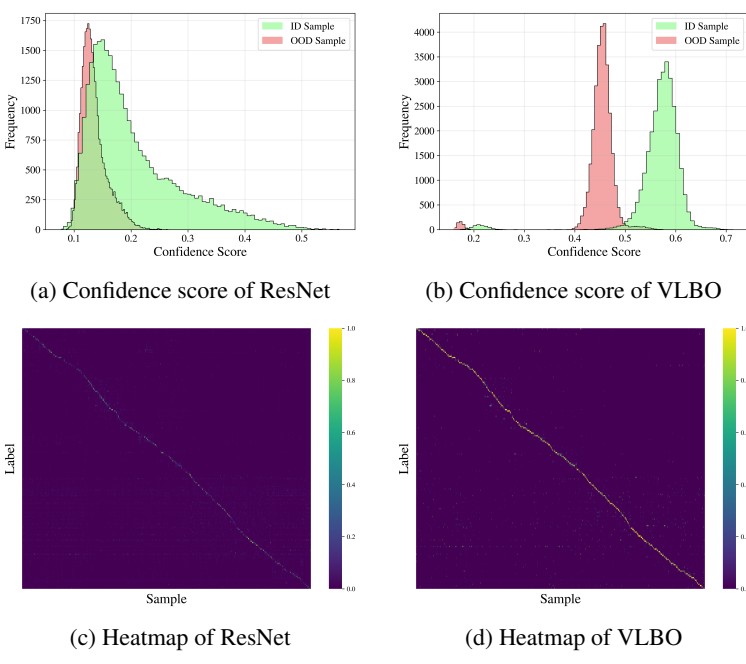

Figure 8: Visualization of OWTTA results on ImageNet-R & MNIST open-world datasets.

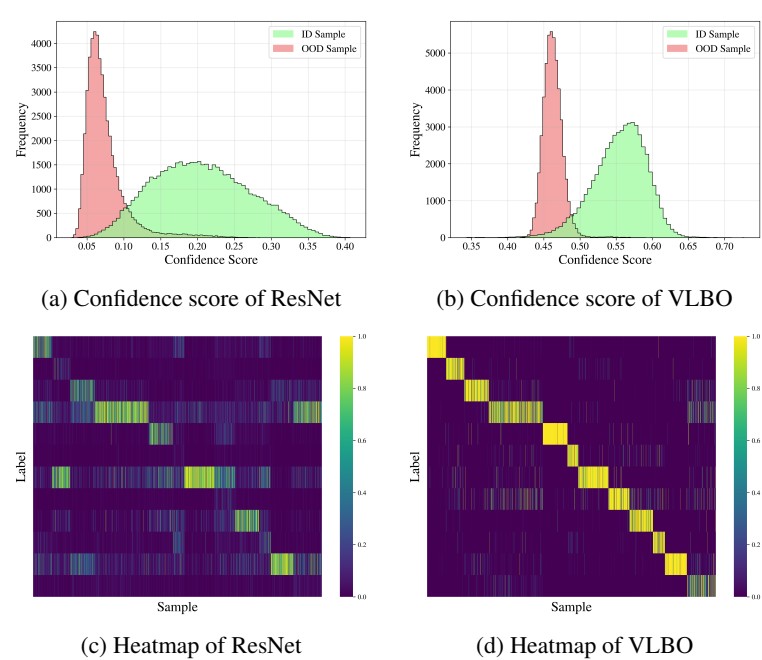

(a) Confidence score of ResNet       (b) Confidence score of VLBO

(c) Heatmap of ResNet       (d) Heatmap of VLBO

Figure 9: Visualization of OWTTA results on VisDA-C & MNIST open-world datasets.

## A.10 ANALYSIS UNDER DIFFERENT OOD RATIOS

In this subsection, we evaluate the robustness of our method under different OOD ratios. Specifically, we vary the OOD ratio $\frac{N_{OOD}}{N_{ID}}$ from 0.1 to 1. The corresponding results, including both $ACC_H$ and standard deviation, are presented in Table 21, Table 22, Table 23, Table 24, and Table 25. We observe slight fluctuations when the OOD ratio becomes extremely low, which is expected since $ACC_O$ is particularly sensitive in the rare-OOD regime where only a few OOD samples are available. Overall, across all datasets and OOD ratios, the proposed method maintains consistently stable performance, demonstrating strong robustness under varying OOD proportions in open-world scenarios.

Table 21: Performance under different OOD ratios on the CIFAR10-C open-world dataset.

| CIFAR10-C | 0.1 | 0.2 | 0.3 | 0.4 | 0.5 | 0.6 | 0.7 | 0.8 | 0.9 | 1 | STD |
|---|---|---|---|---|---|---|---|---|---|---|---|
| Noise | 86.87 | 91.01 | 91.69 | 91.88 | 92.14 | 92.17 | 92.44 | 92.46 | 92.26 | 92.37 | 1.6947 |
| MNIST | 91.13 | 92.33 | 91.39 | 92.65 | 92.59 | 92.93 | 92.38 | 91.27 | 92.47 | 91.79 | 0.6437 |
| SVHN | 85.68 | 90.09 | 90.93 | 90.94 | 91.09 | 91.18 | 91.55 | 91.34 | 91.51 | 89.35 | 1.7814 |
| Tiny | 76.40 | 75.49 | 73.94 | 75.14 | 75.84 | 77.18 | 78.30 | 78.89 | 79.48 | 77.92 | 1.7919 |
| CIFAR100-C | 74.31 | 75.43 | 75.30 | 75.52 | 75.76 | 75.73 | 75.66 | 75.95 | 75.93 | 74.91 | 0.5075 |

Table 22: Performance under different OOD ratios on the CIFAR100-C open-world dataset.

| CIFAR100-C | 0.1 | 0.2 | 0.3 | 0.4 | 0.5 | 0.6 | 0.7 | 0.8 | 0.9 | 1 | STD |
|---|---|---|---|---|---|---|---|---|---|---|---|
| Noise | 60.65 | 63.46 | 64.97 | 64.88 | 68.38 | 64.96 | 67.41 | 63.37 | 64.44 | 69.92 | 2.6900 |
| MNIST | 58.37 | 61.8 | 63.26 | 60.82 | 63.99 | 64.59 | 64.19 | 63.68 | 63.46 | 62.87 | 1.8982 |
| SVHN | 52.75 | 54.7 | 51.87 | 53.97 | 56.4 | 58.38 | 56.3 | 55.89 | 56.81 | 57.56 | 2.0966 |
| Tiny | 47.46 | 48.93 | 48.99 | 49.65 | 50.01 | 51.16 | 51.03 | 52.03 | 52.2 | 51.89 | 1.5831 |
| CIFAR10-C | 45.65 | 46.42 | 47.34 | 47.83 | 48.19 | 48.55 | 48.22 | 48.82 | 49.18 | 48.79 | 1.1283 |

Table 23: Performance under different OOD ratios on the ImageNet-C open-world dataset.

| ImageNet-C | 0.1 | 0.2 | 0.3 | 0.4 | 0.5 | 0.6 | 0.7 | 0.8 | 0.9 | 1 | STD |
|---|---|---|---|---|---|---|---|---|---|---|---|
| Noise | 62.05 | 62.99 | 62.74 | 62.93 | 62.51 | 62.74 | 62.76 | 62.74 | 62.75 | 63.06 | 0.2846 |
| MNIST | 62.90 | 63.08 | 60.01 | 63.62 | 63.75 | 63.68 | 63.72 | 63.61 | 63.87 | 63.60 | 1.1564 |
| SVHN | 62.28 | 62.42 | 62.45 | 62.62 | 62.46 | 62.63 | 62.60 | 62.64 | 62.76 | 62.26 | 0.1645 |

Table 24: Performance under different OOD ratios on the ImageNet-R open-world dataset.

| ImageNet-R | 0.1 | 0.2 | 0.3 | 0.4 | 0.5 | 0.6 | 0.7 | 0.8 | 0.9 | 1 | STD |
|---|---|---|---|---|---|---|---|---|---|---|---|
| Noise | 77.98 | 78.08 | 78.39 | 78.91 | 78.42 | 79.00 | 79.33 | 79.36 | 79.45 | 79.56 | 0.5894 |
| MNIST | 74.78 | 75.88 | 76.00 | 76.36 | 76.14 | 76.63 | 76.59 | 76.96 | 76.95 | 77.53 | 0.7530 |
| SVHN | 76.47 | 77.27 | 77.18 | 77.37 | 77.52 | 77.75 | 77.60 | 78.01 | 77.72 | 77.96 | 0.4502 |

Table 25: Performance under different OOD ratios on the VisDA open-world dataset.

| VisDA | 0.1 | 0.2 | 0.3 | 0.4 | 0.5 | 0.6 | 0.7 | 0.8 | 0.9 | 1 | STD |
|---|---|---|---|---|---|---|---|---|---|---|---|
| Noise | 79.43 | 79.06 | 80.60 | 81.37 | 82.23 | 80.35 | 80.84 | 80.93 | 80.98 | 85.64 | 1.8202 |
| MNIST | 79.78 | 79.66 | 79.32 | 80.31 | 80.62 | 81.10 | 80.89 | 81.53 | 81.96 | 83.16 | 1.1709 |
| SVHN | 84.74 | 85.10 | 86.88 | 86.05 | 85.94 | 85.78 | 85.49 | 84.76 | 86.93 | 87.06 | 0.8725 |

# B  THE USAGE OF LARGE LANGUAGE MODEL

This paper was polished with the assistance of a large language model for language refinement. The research content and conclusions are original contributions of the authors.

