# OpenReview forum: "Empowering Test-Time Adaptation with Complementary Vision-Language Knowledge in Open-World Scenarios"
_ICLR.cc/2026/Conference — ICLR 2026 Conference Withdrawn Submission_

### Official Review · Reviewer_THzq · 2025-10-28

**Soundness:** 2
**Presentation:** 1
**Contribution:** 2
**Rating:** 2
**Confidence:** 4

**Summary:**

The paper proposes a method to address the challenge of open-world test-time adaptation. Conceptually, the approach integrates the vision-language model (VLM) CLIP with a discriminative model (e.g., ResNet) to leverage their complementary strengths. Technically, the proposed framework consists of two key components: 1. Agreement-boosted filtering, which employs VLM guidance to enhance the out-of-distribution (OOD) detection capability of the discriminative model; 2. Semantic-boosted adaptation, which utilizes the semantic representations from the VLM to improve the discriminative model’s adaptation during test time. The effectiveness of the proposed method is validated across multiple standard benchmarks, demonstrating consistent improvements in open-world adaptation performance.

**Strengths:**

- The proposed idea is intuitive. The paper clearly articulates the complementary strengths of **ResNet** and **CLIP** at the conceptual level, followed by two well-defined components that effectively integrate their benefits at the technical level.
- The experimental evaluation is comprehensive, covering both **corruption** and **style-transfer** domain shifts across multiple **out-of-distribution (OOD)** benchmarks. The proposed **VLBO** method consistently outperforms baselines across all datasets and evaluation metrics.

**Weaknesses:**

The paper’s **problem setting and motivation** are not clearly established in the Introduction. The connections between the proposed method, existing approaches, and the broader context of **open-world test-time adaptation (OWTTA)** are vague, making the logical flow difficult to follow. Readers may find the introduction confusing until reaching the Method section, where the framework becomes clearer.

For example, in **Line 44**, the statement __“OWTTA aims to enable models to adapt to the target domain without requiring label data or source statistics”__ may be misleading—it appears to describe domain adaptation, whereas later sections (e.g., **Section 3.1** and **Figure 1**) reveal that OWTTA also addresses **novel-class scenarios**. Additionally, the paper does not clarify how the concepts of **in-distribution (ID)** and **out-of-distribution (OOD)** relate to **domain discrepancy** and **semantic variance**, despite these being central to the problem.

Between **Lines 47–55**, the relationship between **TTA** and **OWTTA** is underexplained. It remains unclear whether OWTTA is a special case of TTA, what their respective challenges are, and how existing TTA methods motivate the proposed OWTTA approach. Similarly, **Lines 56–65** introduce several technical elements (e.g., similarity measures, data filtering, probabilistic formulations) without adequate explanation or conceptual grounding, making it difficult to understand how each component contributes to performance.

To improve readability, the introduction should include a **conceptual overview diagram** summarizing the overall framework and its components. Each component should be introduced following a clear structure: **objective (why)**, **conceptual explanation (what)**, and **technical design (how)**. Overall, the current introduction feels underdeveloped and should be **substantially rewritten** to properly motivate the problem, clarify the conceptual landscape, and prepare readers for the method section.

**Questions:**

- Could the authors clarify the exact problem formulation of OWTTA? Does it focus solely on domain shifts, or also include novel-class scenarios?

- In Line 44, the statement “OWTTA aims to enable models to adapt to the target domain without requiring label data or source statistics” seems to suggest a domain adaptation setting. Could the authors explain how this differs from or extends standard domain adaptation?

- How do the authors define in-distribution (ID) and out-of-distribution (OOD) in the context of this paper? Are these terms related to input distribution, label distribution, or both?

- Is OWTTA a special case or an extension of TTA?

- How do existing TTA methods inspire or motivate the proposed OWTTA approach? Please provide explicit connections or differences.

- Could the authors provide a diagram summarizing the overall framework, highlighting the role and interaction of each component?

---

> ### Author Response · Authors · 2025-11-22
>
> **Q1: Could the authors clarify the exact problem formulation of OWTTA? Does it focus solely on domain shifts, or also include novel-class scenarios?**
>
> **A1:** Thank you for your comments. As described in Figure 1, OWTTA is formulated to address **both** domain shifts and the presence of **novel (unknown) classes**, all under a strict **single-pass test-time constraint**. In this setting, the model must simultaneously adapt to distribution changes and identify OOD samples **without accessing any source statistics (data, labels, or feature distributions).**
>
> **Q2: In Line 44, the statement “OWTTA aims to enable models to adapt to the target domain without requiring label data or source statistics” seems to suggest a domain adaptation setting. Could the authors explain how this differs from or extends standard domain adaptation?**
>
> **A2:** Thank you for your comments. To avoid potential ambiguity, we summarize the differences among the relevant adaptation settings in the following table.
>
>
> | Setting                                | Source Statistics | Domain Shift | Novel Class | Access Pattern |
> | -------------------------------------- | :---------------: | :----------: | :---------: | :------------: |
> | Domain Adaptation                      |        ✔️         |      ✔️      |      ✘      |   Multi-pass   |
> | Source-Free Domain Adaptation          |         ✘         |      ✔️      |      ✘      |   Multi-pass   |
> | Open-Set Source-Free Domain Adaptation |         ✘         |     ✔️        |      ✔️       |   Multi-pass   |
> | Test-Time Domain Adaptation            |         ✘         |      ✔️      |      ✘      |  Single-pass   |
> | Open-World Test-Time Adaptation        |         ✘         |      ✔️      |     ✔️      |  Single-pass   |
>
> *Table 1: The term single-pass means that each sample can be processed only once.*
>
> **Q3: How do the authors define in-distribution (ID) and out-of-distribution (OOD) in the context of this paper? Are these terms related to input distribution, label distribution, or both?**
>
> **A3:** Thank you for your comments. The ID data here refers to **classes** that were **included** during pre-training but whose **distributions differ** at test time. In contrast, OOD data corresponds to unknown classes that **were not seen during pre-training**.
>
> **Q4:** **Is OWTTA a special case or an extension of TTA?**
>
> **A4:** Thank you for your comments. TTA can be viewed as a special case of OWTTA, since it assumes that all test samples belong to known classes and therefore **does not involve novel-class scenarios**.
>
> **Q5: How do existing TTA methods inspire or motivate the proposed OWTTA approach? Please provide explicit connections or differences.**
>
> **A5:**
> **Connection:**  Thank you for your comments. Existing TTA methods typically rely on sharpening the prediction distribution, which is a key mechanism in test-time adaptation. However, such strategies often remain sensitive to distribution shifts, which motivates us to seek a more reliable source of guidance. We observed that CLIP exhibits strong zero-shot generalization and provides **more stable and better predictions** under distribution changes. Inspired by this, we leverage CLIP as a **robust prior** to guide the adaptation process, rather than relying solely on the discriminative model’s own predictions.
>
> **Difference:**
>
> 1. **Setting difference:** OWTTA is a strictly more challenging setting than TTA, as it requires handling both domain shifts and novel classes under a single-pass constraint.
> 2. **Conceptual difference:** Existing methods are either purely discriminative-model based or purely VLM based. In contrast, we introduce a new paradigm that **empowers a discriminative model with VLM knowledge**, providing a principled way to combine their complementary strengths for open-world test-time adaptation.
>
> **Q6:** **Could the authors provide a diagram summarizing the overall framework, highlighting the role and interaction of each component?**
>
> **A6:** Thank you for your comments. In the revised version, we have added a framework diagram that summarizes the overall workflow and highlights the roles and interactions of all components. The diagram has been included in **Fig. 3** to provide a clearer, more intuitive understanding of our proposed method.

---

### Official Review · Reviewer_NEPD · 2025-11-01

**Soundness:** 2
**Presentation:** 3
**Contribution:** 2
**Rating:** 4
**Confidence:** 3

**Summary:**

This paper addresses test-time adaptation in the open-world problem (OWTTA) by leveraging vision-language models (VLMs). Specifically, for improved out-of-distribution (OOD) detection, it employs a discriminative model to filter out OOD samples and further refines this process using VLMs. Then, VLMs are used to enhance the adaptation of discriminative models. Extensive experiments across diverse benchmarks demonstrate the effectiveness of the proposed method.

**Strengths:**

1. The method and motivation of this paper are clearly presented and easy to understand.
2. The observations in OWTTA are quite interesting and form the foundation for the subsequent methods.
3. This OWTTA setting closely resembles real-world scenarios.

**Weaknesses:**

1. In Section 3.2, Observation 1, there is no explicit conclusion stating that samples “tend to be distributed closer to their class prototypes than OOD samples,” yet this statement is used in line 230. Could the authors add this conclusion to Observation 1 to make it easier for readers to comprehend?

2. For table 1, if the model was trained on CIFAR-10 (source) and is tested on MNIST (target), and MNIST has no overlapping classes with CIFAR-10, how is ACC_I (in-distribution accuracy) measured in this case? Could the authors provide more details about this experiment?

3. In the ablation study, for the agreement-boosted filtering module, there is no experiment evaluating the effect of using only the VLM or only the discriminative model to filter OOD samples. Such experiments could demonstrate the effectiveness of combining the two models. Additionally, for the semantic-boosted adaptation module, it would be helpful to explore what happens if the predictions of the two models are simply added rather than multiplied.

4. The paper does not include experiments on time or memory analysis, which are quite important in the OWTTA scenario. Introducing an additional model like the VLM may increase both computational time and memory usage, so it would be helpful to evaluate these costs.

**Questions:**

Please see the above weaknesses. If you can conduct additional experiments to further evaluate your method, I would be willing to raise my score.

---

> ### Author Response · Authors · 2025-11-22
>
> **Q1:In Section 3.2, Observation 1, there is no explicit conclusion stating that samples “tend to be distributed closer to their class prototypes than OOD samples,” yet this statement is used in line 230. Could the authors add this conclusion to Observation 1 to make it easier for readers to comprehend?**
>
> **A1:** Thank you for pointing this out. In the revised version, we have updated Observation 1 in line 197 to explicitly include this conclusion, thereby clarifying the connection and improving readability for the readers. For your convenience, we copy the revised text below:
> "As shown in Fig. 2a, the learned representations of ID samples cluster around the classifier weights, which serve as prototypes of the source classes. In contrast, OOD samples are located far from all prototypes."
>
> **Q2: For table 1, if the model was trained on CIFAR-10 (source) and is tested on MNIST (target), and MNIST has no overlapping classes with CIFAR-10, how is ACC_I (in-distribution accuracy) measured in this case? Could the authors provide more details about this experiment?**
>
> **A2:** Thank you for your comments. In OWTTA, we evaluate performance using two metrics: the in-distribution classification accuracy ($ACC_I$) and the OOD detection accuracy ($ACC_O$). In our experimental setup, $ACC_I$ is computed as the **number of correctly classified ID samples divided by the total number of ID samples**. Therefore, in Table 1, $ACC_I$ is _not_ computed on MNIST, since MNIST serves solely as the OOD dataset in this setting.
> Conversely, MNIST is evaluated through $ACC_O$, which measures the proportion of MNIST samples that are correctly identified as OOD.

---

> ### Author Response · Authors · 2025-11-22
>
> **Q3: In the ablation study, for the agreement-boosted filtering module, there is no experiment evaluating the effect of using only the VLM or only the discriminative model to filter OOD samples. Such experiments could demonstrate the effectiveness of combining the two models. Additionally, for the semantic-boosted adaptation module, it would be helpful to explore what happens if the predictions of the two models are simply added rather than multiplied.**
>
> **A3:** Thank you for your insightful comments. Regarding your first question, using only the VLM corresponds to the CLIP baseline, while using only the discriminative model corresponds to the second row of the ablation tables. For your convenience, we restate their $ACC_O$ results in the tables below.
>
> | **CIFAR10-C** | Noise | MNIST | SVHN | Tiny | CIFAR100-C |
> | :-----------------: | :----: | :---: | :---: | :---: | :--------: |
> |       ResNet       | 100.00 | 97.76 | 83.65 | 81.07 |   76.39   |
> |        CLIP        | 99.97 | 99.69 | 96.70 | 81.27 |   75.68   |
> |        Ours        | 100.00 | 97.52 | 95.45 | 84.76 |   78.78   |
>
> | **CIFAR100-C** | Noise | MNIST | SVHN | Tiny | CIFAR10-C |
> | -------------------- | :---: | :---: | :---: | :---: | :-------: |
> | ResNet               | 99.96 | 94.29 | 77.20 | 72.97 |   76.41   |
> | CLIP                 | 2.99 | 99.29 | 94.88 | 66.30 |   68.01   |
> | Ours                 | 99.91 | 95.68 | 81.66 | 74.37 |   76.02   |
>
> | **ImageNet-C** | Noise | MNIST | SVHN |
> | -------------------- | :----: | :---: | :---: |
> | ResNet               | 73.23 | 73.63 | 74.32 |
> | CLIP                 | 100.00 | 99.96 | 96.39 |
> | Ours                 | 100.00 | 99.00 | 98.20 |
>
> | **ImageNet-R** | Noise | MNIST | SVHN |
> | -------------------- | :----: | :---: | :---: |
> | ResNet               | 98.71 | 71.50 | 95.61 |
> | CLIP                 | 100.00 | 99.85 | 99.16 |
> | Ours                 | 99.04 | 97.63 | 97.76 |
>
> | **VisDA** | Noise | MNIST | SVHN |
> | --------------- | :----: | :---: | :---: |
> | ResNet          | 99.41 | 99.17 | 97.88 |
> | CLIP            | 100.00 | 99.71 | 97.35 |
> | Ours            | 99.63 | 99.49 | 98.70 |
>
> For your second question, we replace the PoE formulation in Eq. (12) with a simple additive fusion, where the logits from the VLM and the discriminative model are added together. The corresponding results in terms of $ACC_H$  are shown in the tables below.
>
> | CIFAR10-C | Noise | MNIST | SVHN | Tiny | CIFAR100-C |
> | :-------: | :---: | :---: | :---: | :---: | :--------: |
> |    Add    | 92.18 | 91.37 | 89.39 | 77.16 |   74.64   |
> |   VLBO   | 92.37 | 91.79 | 89.35 | 77.29 |   74.91   |
>
> | **CIFAR100-C** | **Noise** | **MNIST** | **SVHN** | **Tiny** | **CIFAR10-C** |
> | :------------------: | :-------------: | :-------------: | :------------: | :------------: | :------------------: |
> |         Add         |      65.47      |      60.88      |     57.41     |     51.00     |        49.31        |
> |         VLBO         |      69.92      |      62.87      |     57.56     |     51.89     |        48.79        |
>
> | **ImageNet-C** | **Noise** | **MNIST** | **SVHN** |
> | :------------------: | :-------------: | :-------------: | :------------: |
> |         Add         |      59.28      |      60.02      |     58.71     |
> |         VLBO         |      63.03      |      63.60      |     62.26     |
>
> | **ImageNet-R** | **Noise** | **MNIST** | **SVHN** |
> | :------------------: | :-------------: | :-------------: | :------------: |
> |         Add         |      80.10      |      77.32      |     77.63     |
> |         VLBO         |      79.56      |      77.53      |     77.96     |
>
> | **VisDA** | **Noise** | **MNIST** | **SVHN** |
> | :-------------: | :-------------: | :-------------: | :------------: |
> |       Add       |      85.19      |      83.05      |     86.73     |
> |      VLBO      |      85.64      |      83.16      |     87.06     |

---

> > ### Author Response · Authors · 2025-11-22
> >
> > **Q4: The paper does not include experiments on time or memory analysis, which are quite important in the OWTTA scenario. Introducing an additional model like the VLM may increase both computational time and memory usage, so it would be helpful to evaluate these costs.**
> >
> > **A4:**  Thank you for highlighting this detail. We have evaluated the wall-clock time (s) and average memory consumption (MB) on all open-world datasets. The results are reported in Section 5.4 (Table 7) and in Appendix A.8 (Tables 17–20) of the revised manuscript. For the reviewer’s convenience, we reproduce below the **subset** of results corresponding to the setting with MNIST as the OOD samples. From these results, we can conclude that VLBO achieves a reasonable balance between computational overhead and adaptation performance.
> >
> > | CIFAR10-C |  Time  | Memory | $ACC_H$ | CIFAR100-C |  Time  | Memory | $ACC_H$ | ImageNet-C |  Time  | Memory | $ACC_H$ |
> > | :-------: | :----: | :-----: | :-------: | :----------: | :----: | :-----: | :-------: | :--------: | :-----: | :-----: | :-------: |
> > |   CLIP   | 0.1611 | 507.26 |   76.69   |    CLIP    | 0.1569 | 508.36 |   47.41   |    CLIP    | 0.1965 | 595.91 |   46.08   |
> > |   C-TPT   | 0.4789 | 1220.96 |   76.37   |   C-TPT   | 0.4925 | 1235.68 |   35.12   |   C-TPT   | 1.2929 | 1387.16 |   21.19   |
> > |   CLIPN   | 0.3184 | 403.25 |   32.42   |   CLIPN   | 0.3154 | 403.69 |   21.91   |   CLIPN   | 0.3054 | 408.46 |   14.11   |
> > | NegSample | 0.1130 | 578.93 |   62.53   | NegSample | 0.1147 | 579.33 |   43.34   | NegSample | 0.1246 | 598.33 |   44.05   |
> > |   WATT   | 8.8878 | 1099.38 |   76.63   |    WATT    | 7.0037 | 1099.60 |   49.82   |    WATT    | 18.3008 | 1102.08 |   47.31   |
> > |   Ours   | 0.6569 | 782.32 |   91.79   |    Ours    | 0.5425 | 912.22 |   62.87   |    Ours    | 0.9591 | 1337.59 |   63.60   |
> >
> > | ImageNet-R |  Time  | Memory | $ACC_H$ |   VisDA   |  Time  | Memory | $ACC_H$ |
> > | :--------: | :----: | :-----: | :-------: | :-------: | :------: | :-----: | :-------: |
> > |    CLIP    | 0.1877 | 593.34 |   71.05   |   CLIP   | 0.1913 | 580.69 |   74.77   |
> > |   C-TPT   | 0.5438 | 1260.39 |   68.95   |   C-TPT   | 0.4686 | 1221.33 |   75.04   |
> > |   CLIPN   | 0.3079 | 404.49 |   31.41   |   CLIPN   | 0.3041 | 403.56 |   33.37   |
> > | NegSample | 0.1195 | 596.75 |   76.39   | NegSample | 0.1143 | 596.27 |   56.63   |
> > |    WATT    | 9.0740 | 1100.16 |   70.80   |   WATT   | 8.8650 | 1099.68 |   74.86   |
> > |    Ours    | 0.9742 | 1172.48 |   77.53   |   Ours   | 0.6106 | 770.25 |   83.16   |

---

### Official Review · Reviewer_JY9f · 2025-11-01

**Soundness:** 2
**Presentation:** 3
**Contribution:** 2
**Rating:** 2
**Confidence:** 4

**Summary:**

This paper introduces VLBO (Vision-Language knowledge Boosted Open-world Test-Time Adaptation), which integrates a discriminative model (ResNet) and a vision-language model (CLIP) for open-world test-time adaptation (OWTTA).
The method contains two modules for two phases (OOD detection and TTA):
(1) Agreement-Boosted Filtering (AF), which fuses the confidence of ResNet and CLIP to filter OOD samples;
(2) Semantic-Boosted Adaptation (SA), which fuses the logits of both models via a product-of-experts formulation to adapt the ResNet.
Experiments under the OWTTA setup show that VLBO achieves better performance than ResNet-based TTA baselines and CLIP-based OOD detection baselines.

**Strengths:**

1. The work addresses the practical open-world test-time adaptation setting, which combines OOD detection and TTA.
2. The ablation study is clear and isolates the contribution of each module.

**Weaknesses:**

1. "Agreement-Boosted Filtering" and "Semantic-Boosted Adaptation" correspond to simple confidence score averaging and logit summation; there seems to be little theoretical or algorithmic innovation.
2. The empirical “Observations 1–3” (e.g., ResNet encodes task-specific features, CLIP is more robust to distribution shifts) are already well-known and serve more as narrative justification than new insights.
3. OWTTA can be viewed as TTA + OOD detection. VLBO includes two separate parts designed for OOD detection and TTA phases, so the comparison with TTA baselines and OOD detection baselines appears not entirely fair. Would a SoTA OOD detection method combined with a SoTA TTA method be a fairer baseline?
4. There seems to be no discussion of computational overhead or latency, which matters in test-time settings.

**Questions:**

1. To my understanding, OWTTA here is TTA plus OOD detection. There have been two lines of research exploring the two problems separately. To address OWTTA, one can simply combine an OOD detection and a TTA method as two sequential phases for each test step. Would the authors agree with this point? As mentioned in point 3 of Weaknesses, would a SoTA OOD detection method plus a SoTA TTA method be a fairer baseline?
2. The paper does not appear to clearly explain how the baselines are evaluated under the OWTTA setting. OOD detection baselines are designed for detection and do not seem to perform adaptation. It is unclear whether they are re-implemented with an adaptation phase, or simply applied in a frozen zero-shot manner.

---

> ### Author Response · Authors · 2025-11-22
>
> **Q1: "Agreement-Boosted Filtering" and "Semantic-Boosted Adaptation" correspond to simple confidence score averaging and logit summation; there seems to be little theoretical or algorithmic innovation.**
>
> **A1:** We respectfully disagree with the reviewer’s comment.
>
> Although the final formulation reduces to weighted confidence scores and logit summation, both Agreement-Boosted Filtering (AF) and Semantic-Boosted Adaptation (SA) are derived from a probabilistic framework (**Eq. (1), Eq. (5), Eq. (6), Eq. (12) in the manuscript).** Note that the probabilistic formulation for confidence-based OOD filtering   and the logit summation derived from the joint probabilitics of VLMs and task-specific discrminative models are first proposed for the OWTTA task.
>
> What's more, even in its simple form, the agreement-boosted filtering makes OOD samples more distinguishable, and semantic-boosted adaptation  corrects false predictions by the VLM and the task-specific discriminative model through their agreement on the input image semantics, as detailed explained in our newly included diagram (**Fig. 3**). These advances have been demonstrated by empirical results (Fig. 5-Fig. 9). We argue that the benefits demonstrated by VLBO, even with these simple implementations, should be recognized as a valuable contribution to this domain.
>
> **Q2: The empirical “Observations 1–3” (e.g., ResNet encodes task-specific features, CLIP is more robust to distribution shifts) are already well-known and serve more as narrative justification than new insights.**
>
> **A2:** Thank you for pointing this out.
> We agree that the empirical phenomena summarized in Observations 1–3 are well-known individually. They are **not** presented as standalone contributions, but rather as a way to surface the **implicit structural clues** that directly motivate the design of VLBO.
>
> Although these properties are familiar on their own, their **joint implications for OWTTA have not been described in prior work.** When considered together, they clarify how the strengths of a VLM and a discriminative model **complement each other** and why an informed integration of the two is needed in the open-world single-pass setting. These observations form the empirical basis for our conceptual framework, which transfers VLM priors to enhance discriminative adaptation.
>
> In this sense, the observations are not merely narrative justification.
> They bring to light the underlying semantic and representational mismatch that any effective OWTTA method must resolve. This connection has not been identified or linked together in existing studies. Recognizing this connection and building a working framework upon it is an essential part of our contribution.
>
> **Q3: OWTTA can be viewed as TTA + OOD detection. VLBO includes two separate parts designed for OOD detection and TTA phases, so the comparison with TTA baselines and OOD detection baselines appears not entirely fair. Would a SoTA OOD detection method combined with a SoTA TTA method be a fairer baseline?**
>
> **A3:** Thank you for your comments. Our choice of baselines follows the reference paper “On the Robustness of Open-World Test-Time Training: Self-Training with Dynamic Prototype Expansion” (ICCV 2023), which established the evaluation protocol for OWTTA.
> Combining a SoTA OOD detection method with a SoTA TTA method does not constitute a fair comparison to OWTTA methods, as OWTTA is designed as a unified framework that jointly handles both domain discrepancy and semantic variance, rather than treating them as two independent stages.

---

> ### Author Response · Authors · 2025-11-22
>
> **Q4: There seems to be no discussion of computational overhead or latency, which matters in test-time settings.**
>
> **A4:** Thank you for raising this point. In the revised manuscript, we have added a detailed analysis of running time (s) and average memory cost (MB), which is provided in Section 5.4 and Appendix A.8. For the reviewer’s convenience, we present below the results corresponding to the setting with MNIST as the OOD dataset.
>
> | CIFAR10-C |  Time  | Memory | $ACC_H$ | CIFAR100-C |  Time  | Memory | $ACC_H$ | ImageNet-C |  Time  | Memory | $ACC_H$ |
> | :-------: | :----: | :-----: | :-------: | :--------: | :------: | :-------: | :-------: | :--------: | :------: | :------: | :--------: |
> |   CLIP   | 0.1611 | 507.26 |   76.69   |    CLIP    | 0.1569 | 508.36 |   47.41   |    CLIP    | 0.1965 | 595.91 |   46.08   |
> |   C-TPT   | 0.4789 | 1220.96 |   76.37   |   C-TPT   | 0.4925 | 1235.68 |   35.12   |   C-TPT   | 1.2929 | 1387.16 |   21.19   |
> |   CLIPN   | 0.3184 | 403.25 |   32.42   |   CLIPN   | 0.3154 | 403.69 |   21.91   |   CLIPN   | 0.3054 | 408.46 |   14.11   |
> | NegSample | 0.1130 | 578.93 |   62.53   | NegSample | 0.1147 | 579.33 |   43.34   | NegSample | 0.1246 | 598.33 |   44.05   |
> |   WATT   | 8.8878 | 1099.38 |   76.63   |    WATT    | 7.0037 | 1099.60 |   49.82   |    WATT    | 18.3008 | 1102.08 |   47.31   |
> |   Ours   | 0.6569 | 782.32 |   91.79   |    Ours    | 0.5425 | 912.22 |   62.87   |    Ours    | 0.9591 | 1337.59 |   63.60   |
>
>
> | ImageNet-R |  Time  | Memory | $ACC_H$ |   VisDA   |  Time  | Memory | $ACC_H$ |
> | :--------: | :----: | :-----: | :-------: | :-------: | :----: | :-----: | :-------: |
> |    CLIP    | 0.1877 | 593.34 |   71.05   |   CLIP   | 0.1913 | 580.69 |   74.77   |
> |   C-TPT   | 0.5438 | 1260.39 |   68.95   |   C-TPT   | 0.4686 | 1221.33 |   75.04   |
> |   CLIPN   | 0.3079 | 404.49 |   31.41   |   CLIPN   | 0.3041 | 403.56 |   33.37   |
> | NegSample | 0.1195 | 596.75 |   76.39   | NegSample | 0.1143 | 596.27 |   56.63   |
> |    WATT    | 9.0740 | 1100.16 |   70.80   |   WATT   | 8.8650 | 1099.68 |   74.86   |
> |    Ours    | 0.9742 | 1172.48 |   77.53   |   Ours   | 0.6106 | 770.25 |   83.16   |
>
> These results show that VLBO achieves a more favorable balance between performance and computational overhead compared with the evaluated baselines.
>
> **Q5: To my understanding, OWTTA here is TTA plus OOD detection. There have been two lines of research exploring the two problems separately. To address OWTTA, one can simply combine an OOD detection and a TTA method as two sequential phases for each test step. Would the authors agree with this point? As mentioned in point 3 of Weaknesses, would a SoTA OOD detection method plus a SoTA TTA method be a fairer baseline?**
>
> **A5:**  We appreciate the reviewer’s comment. While OWTTA indeed involves both distribution shifts and the presence of unknown samples, we would like to clarify that it is **not** a simple “TTA + OOD detection’’ problem.
>
> OOD detection and TTA are designed with **different and often conflicting objectives**.
>
> - **OOD detection** aims only to separate unknown samples, and does **not** consider maintaining ID classification accuracy.
> - **TTA methods**, on the other hand, assume that all incoming samples are ID and focus on sharpening the logits or increasing confidence.
>
> When unknown samples appear, a TTA module that assumes full-ID input will be **misguided by OOD features**, leading to degraded performance. Likewise, an OOD detector operating independently of the adaptation process may generate incompatible decisions that disrupt adaptation. Therefore, simply combining two methods as sequential phases does not form a coherent solution to OWTTA.
>
> For this reason, we argue that combining a SoTA OOD detector with a SoTA TTA method does not constitute a fair or meaningful baseline. This is also consistent with the OWTTA evaluation protocol adopted in prior work, which treats OWTTA as a unified problem rather than two separable stages.
>
> **Q6: The paper does not appear to clearly explain how the baselines are evaluated under the OWTTA setting. OOD detection baselines are designed for detection and do not seem to perform adaptation. It is unclear whether they are re-implemented with an adaptation phase, or simply applied in a frozen zero-shot manner.**
>
> **A6:** We appreciate the reviewer’s remarks. In the revised version, we have clarified the implementation protocol for evaluating the baselines under the OWTTA setting in Appendix A.4 (line 826). For the reviewer’s convenience, we restate the clarification below:
>
> “CLIP-based OOD detection methods are evaluated in their original form. We report their OOD detection performance directly and compute the ID classification accuracy using either the corresponding trained encoder or the zero-shot CLIP model with updated prompts, depending on the design of each method.”

---

> > ### Comment · Reviewer_JY9f · 2025-11-27
> >
> > Thanks for the authors' rebuttal.
> >
> > **Regarding "Probabilistic Formulation" (Q1):** I appreciate the clarification, but I maintain that the term "probabilistic formulation" is an overstatement. As the authors admit, the method effectively reduces to weighted averaging (Eq. 9) and product-of-logits fusion (Eq. 12). The derivation provides a notational wrapper, but there is no genuine generative modeling, prior/posterior integration, or uncertainty quantification involved. It remains a deterministic fusion of confidence scores and logits, which is standard ensembling, not a novel probabilistic modeling contribution.
> >
> >
> > **Regarding the Baseline and OWTTA Logic (Q3/Q5):** I find the argument in A5 to be logically inconsistent, particularly given the structure of the proposed VLBO method. The authors argue that a TTA module in a sequential pipeline would be "misguided by OOD features." However, the purpose of the preceding OOD detector is precisely to filter those features out before adaptation occurs. If the detector works, the TTA module sees only "ID" data, resolving the conflict the authors describe.
> >
> > More importantly, Algorithm 1 of the paper explicitly follows this exact sequential logic: First, "Filter out OOD samples" (The Detection Phase), then "Semantic-boosted adaptation" (The Adaptation Phase). Since the proposed method itself relies on a "Filter $\rightarrow$ Adapt" sequence, it is contradictory to claim that a baseline composed of "SoTA Filter $\rightarrow$ SoTA Adapter" is an unfair or incoherent comparison. The refusal to compare against this straightforward baseline significantly weakens the empirical validation of the proposed unified framework.

---

### Official Review · Reviewer_GuQx · 2025-11-01

**Soundness:** 2
**Presentation:** 2
**Contribution:** 2
**Rating:** 2
**Confidence:** 4

**Summary:**

This paper introduces VLBO, an open-world test-time adaptation (OWTTA) framework that leverages a vision-language model (i.e., CLIP) to enhance model robustness under both domain discrepancy and semantic variance. VLBO comprises two key components: (1) Agreement-boosted Filtering (AF), which identifies and filters out out-of-distribution (OOD) samples by combining the confidence of a discriminative model (ResNet) with image–text similarity cues from CLIP; and
(2) Semantic-boosted Adaptation (SA), which adapts the discriminative model using CLIP’s semantic priors through a product-of-experts formulation. Experiments on five open-world benchmarks (i.e., CIFAR10-C, CIFAR100-C, ImageNet-C, ImageNet-R, and VisDA-C) demonstrate consistent performance gains over both discriminative and vision-language model baselines.

**Strengths:**

+ OWTTA represents an important and practical research challenge. This paper is one of several concurrent studies that demonstrate the effectiveness of incorporating CLIP into OWTTA settings.

**Weaknesses:**

- The novelty of the proposed method appears limited. Specifically, the AF module is essentially a straightforward extension of conventional confidence-based OOD filtering, where CLIP-based similarity is incorporated as an auxiliary confidence measure. Similarly, the SA module largely follows standard domain adaptation techniques based on pseudo-labeling of target data. Overall, the method does not introduce a fundamentally new learning principle or a theoretically grounded formulation.
- As discussed in §2, several prior studies have already explored CLIP-guided OOD detection and adaptation. However, the paper does not clearly articulate the motivation for VLBO or its conceptual distinctions from these existing approaches. Moreover, some recent relevant methods (e.g., [1-2]) are missing from the experimental comparisons (§5, Tables 1-4).
- Algorithm 1 appears oversimplified. For example, the definition and role of T are unclear: does it refer to adaptation steps, test-time batches, or epochs?
- The relationship between OWTTA and Open-Set Source-Free Domain Adaptation (OS-SFDA) (e.g., [3]) is not clearly discussed. Although these two settings may not be identical, they are closely related; both address adaptation under distribution shifts with the presence of unknown target classes. It would be beneficial for the paper to explicitly clarify their conceptual differences and connections in §2. In addition, including comparisons with representative OS-SFDA methods in §5 would provide a more comprehensive evaluation of the proposed approach.

[1] C. Cao et al., Noisy Test-Time Adaptation in Vision-Language Models, in ICLR, 2025.

[2] D. Osowiechi et al., WATT: weight average test time adaptation of CLIP, in NeurIPS, 2024.

[3] F. Wan et al., Unveiling the Unknown: Unleashing the Power of Unknown to Known in Open-Set Source-Free Domain Adaptation, in CVPR, 2024.

**Questions:**

Please refer to the Weakness section.

---

> ### Author Response · Authors · 2025-11-22
>
> **Q1: The novelty of the proposed method appears limited. Specifically, the AF module is essentially a straightforward extension of conventional confidence-based OOD filtering, where CLIP-based similarity is incorporated as an auxiliary confidence measure. Similarly, the SA module largely follows standard domain adaptation techniques based on pseudo-labeling of target data. Overall, the method does not introduce a fundamentally new learning principle or a theoretically grounded formulation.**
>
> **A1:** We respectfully disagree with the reviewer. It appears there may be a misunderstanding regarding the design of our framework. Specifically,
>
> * AF is not a trivial extension of conventional confidence-base OOD filtering. We have two technical innovations: (1) we establish a probabilistic formulation for confidence-based OOD filtering (**Eq. (5), Eq. (6)** in the manuscript), which has never been proposed in the literature. (2) With the probabilistic framework, we design a weighted strategy (**Eq. (9)**) to augments the discriminative model’s confidence with semantic cues from CLIP.
> * Standard domain adaptation techniques based on pseudo-labeling of target data is not related to the design of our SA module. SA is mainly designed to integrate the signal from VLMs and task-specific discriminative models, which also relies on a probabilistic formulation (**Eq. (12)**). In contrast, the mentioned adaption techniques just relies on the task-spectifc discriminative model, not involving any integration of two types of models.
>
> Given the literature gap in how vision–language foundation models can effectively guide task-specific discriminative models in OWTTA, VLBO offers a unified and effective framework that leverages the complementary strengths of these two model types, thereby establishing a new foundational pipeline for OWTTA.

---

> ### Author Response · Authors · 2025-11-22
>
> **Q2: As discussed in §2, several prior studies have already explored CLIP-guided OOD detection and adaptation. However, the paper does not clearly articulate the motivation for VLBO or its conceptual distinctions from these existing approaches. Moreover, some recent relevant methods (e.g., [1-2]) are missing from the experimental comparisons (§5, Tables 1-4).**
>
> **A2:** Thank you for your comments.
> Regarding the first point, Below we clarify both the motivation of VLBO and its conceptual distinctions from existing CLIP-guided OOD detection and adaptation methods.
>
> 1. **Motivation of VLBO**
>    VLBO is motivated by the need to **combine the complementary strengths of a large VLM and a lightweight discriminative model** within a single test-time framework. Large models provide strong general semantic priors but may fail on domain-specific tasks or under distribution shifts, while small discriminative models offer task-specialized decision boundaries but lack robustness. VLBO aims to **leverage the informative priors encoded in the VLM** while **preserving the efficiency and task specificity of the discriminative model**, yielding a method that benefits from both generalization and discriminative power.
> 2. **Conceptual distinctions from prior work (please see Table 1 in the response to all reviewers)**
>
>    (1) **Beyond OOD detection:** VLBO not only identifies distributional anomalies but **performs test-time adaptation** to correct distribution shifts, whereas CLIP-based OOD detection methods lack this capability.
>
>    (2) **Beyond adaptation-only methods:** CLIP-guided adaptation methods assume all test samples are in-distribution and thus fail when OOD inputs appear. In contrast, **VLBO jointly performs adaptation and reliable OOD identification**, preventing harmful updates.
>
>    (3) **Unified framework:** VLBO is a unified **test-time** framework that supports both detection and adaptation, whereas existing methods typically specialize in only one of these tasks. This unified design allows VLBO to handle realistic deployment scenarios where test-time data may contain both data distribution shifts and unexpected OOD inputs.
>
> For the second question, we have included additional experiments with [2] in the revised version to further validate the effectiveness of VLBO.
>
> In our comparison protocol, we consider methods to be suitable baselines only if they **modify or adapt the embedding space**, since this is essential for the OWTTA setting. Methods that do not modify the embedding space are generally limited by the native behavior of CLIP and therefore cannot serve as meaningful adaptation baselines.
>
> Regarding [1] (AdaND), it is an OOD detection method, not targetting OWTTA. It does not have ID adaption process, namely, it does not modify the embedding space and therefore operates fundamentally differently from the methods considered in our comparison. So, AdaND is not suitable for comparison in our setting.
>
> For WATT [2], we have conducted experiments on all datasets, and the results are reported in Tables 1–4 of the revised manuscript, highlighted in blue. For the reviewer’s convenience, a subset of the results ($ACC_H$) is also provided below.
>
> | CIFAR10-C | Noise | MNIST | SVHN | Tiny | CIFAR100-C |
> | :-------: | :---: | :---: | :---: | :---: | :--------: |
> |   WATT   | 73.00 | 76.63 | 78.94 | 76.75 |   70.34   |
> |   VLBO   | 92.37 | 91.79 | 89.35 | 77.29 |   74.91   |
>
> | **CIFAR100-C** | **Noise** | **MNIST** | **SVHN** | **Tiny** | **CIFAR10-C** |
> | :------------------: | :-------------: | :-------------: | :------------: | :------------: | :------------------: |
> |         WATT         |      5.53      |      49.82      |     49.27     |     50.96     |        44.63        |
> |         VLBO         |      69.92      |      62.87      |     57.56     |     51.89     |        48.79        |
>
> | **ImageNet-C** | **Noise** | **MNIST** | **SVHN** |
> | :------------------: | :-------------: | :-------------: | :------------: |
> |         WATT         |      44.64      |      47.31      |     46.69     |
> |         VLBO         |      63.03      |      63.60      |     62.26     |
>
> | **ImageNet-R** | **Noise** | **MNIST** | **SVHN** |
> | :------------------: | :-------------: | :-------------: | :------------: |
> |         WATT         |      62.93      |      70.80      |     75.27     |
> |         VLBO         |      79.56      |      77.53      |     77.96     |
>
> | **VisDA** | **Noise** | **MNIST** | **SVHN** |
> | :-------------: | :-------------: | :-------------: | :------------: |
> |      WATT      |      65.41      |      74.86      |     74.29     |
> |      VLBO      |      85.64      |      83.16      |     87.06     |

---

> ### Author Response · Authors · 2025-11-22
>
> **Q3: Algorithm 1 appears oversimplified. For example, the definition and role of T are unclear: does it refer to adaptation steps, test-time batches, or epochs?**
>
> **A3:** Thank you for pointing this out. In Algorithm 1, _T_ denotes the total time index, i.e., the number of test-time batches to be processed. We have clarified this definition in the revised version to avoid ambiguity (see line 338).
>
> **Q4: The relationship between OWTTA and Open-Set Source-Free Domain Adaptation (OS-SFDA) (e.g., [3]) is not clearly discussed. Although these two settings may not be identical, they are closely related; both address adaptation under distribution shifts with the presence of unknown target classes. It would be beneficial for the paper to explicitly clarify their conceptual differences and connections in §2. In addition, including comparisons with representative OS-SFDA methods in §5 would provide a more comprehensive evaluation of the proposed approach.**
>
> **A4:** Thank you for your comments.
> Concerning your first question, while OWTTA and OS-SFDA both involve distribution shift with unknown classes, OS-SFDA performs offline, multi-pass adaptation on the full target set, whereas OWTTA is strictly online and single-pass.  The contrasts among these settings are made more explicit in the table below.
>
> | Setting                                | Source Statistics | Domain Shift | Novel Class | Access Pattern |
> | -------------------------------------- | :---------------: | :----------: | :---------: | :------------: |
> | Domain Adaptation                      |        ✔️         |      ✔️      |      ✘      |   Multi-pass   |
> | Source-Free Domain Adaptation          |         ✘         |      ✔️      |      ✘      |   Multi-pass   |
> | Open-Set Source-Free Domain Adaptation |         ✘         |      ✔️       |      ✔️      |   Multi-pass   |
> | Test-Time Domain Adaptation            |         ✘         |      ✔️      |      ✘      |  Single-pass   |
> | Open-World Test-Time Adaptation        |         ✘         |      ✔️      |     ✔️      |  Single-pass   |
>
> *The term single-pass means that each sample can be processed only once.*
>
> In the revised version, we have added a clarification in Section 2, line 98 to more explicitly articulate these conceptual differences for readers. For your convenience, we include the revised passage below:
> "Open-Set Domain adaptation. Compared with open-world test-time adaptation, open-set domain adaptation (OSDA) (Busto & Gall, 2017; Pham et al., 2025; Choe et al., 2024) assumes access to the entire target batch during adaptation and allows the target domain to contain novel categories absent from the source. Recent advances in open-set source-free domain adaptation (OS-SFDA) (Yu et al., 2025; Liu et al., 2025; Wan et al., 2024) further enable adaptation without accessing source data or source statistics. However, under limited target data accessibility, these techniques may face challenges in fully satisfying the needs of open-world scenarios."
>
> As for the second question, we note that OS-SFDA methods require **full access** to the target data, typically assuming that the entire target batch is available during adaptation. This assumption fundamentally differs from OWTTA, which enforces a strict single-pass, streaming setting without access to future data. For this reason, OS-SFDA methods cannot be fairly evaluated under our constraints and are therefore not included in the comparison.

---

### Official Review · Reviewer_fTEd · 2025-11-05

**Soundness:** 3
**Presentation:** 3
**Contribution:** 3
**Rating:** 6
**Confidence:** 3

**Summary:**

The paper tackles open-world test-time adaptation where both domain discrepancy and semantic variance appear at inference. It proposes VLBO, a framework that augments a discriminative backbone with vision–language knowledge from CLIP in two stages. Experiments on different datasets show consistent gains in the harmonic mean ACCH over strong TTA and CLIP baselines.

**Strengths:**

- Well-motivated decomposition (AF + SA): AF leverages discriminative, task-specific geometry while SA imports transferable semantics via PoE; this matches their observations and improves robustness.
- Clear probabilistic grounding for AF: mixture-of-Gaussians view with a practical k-means limit on 1-D confidence; simple and stable.
- Effective and consistent results: Notable ACCH gains on ImageNet-C/R and VisDA-C, outperforming TTA and CLIP baselines
- Solid training objective: combining pseudo-label CE, prototype MSE, and diversity regularization is a reasonable recipe for stable online adaptation.

**Weaknesses:**

- Open-world sets fix an equal number of ID and OOD examples; real streams often have skewed OOD rates. Please report robustness across OOD ratios and streaming orders.
- VLBO runs both ResNet and CLIP per test sample plus clustering. Please report wall-clock and memory overhead vs. other baselines

**Questions:**

- Missing reference on agreement-boosted between domains https://arxiv.org/abs/2406.09353. They also use confidence-based filtering for generating pseudo labels
- How does VLBO behave when OOD prevalence varies?

---

> ### Author Response · Authors · 2025-11-22
>
> **Q1: Open-world sets fix an equal number of ID and OOD examples; real streams often have skewed OOD rates. Please report robustness across OOD ratios and streaming orders.**
>
> **A1:** Thank you for this constructive comment.
> We have evaluated the robustness of the proposed method under different OOD ratios $\frac{N_{\mathrm{OOD}}}{N_{\mathrm{ID}}}$, ranging from 0.1 to 1.0. The corresponding results, including both $ACC_H$ and standard deviation, are presented in the tables below.
> It is worth noting that certain datasets exhibit slight fluctuations when the OOD ratio becomes extremely low. This behavior is expected, as $ACC_O$ is inherently sensitive in the rare-OOD regime where only a very limited number of OOD samples are available.
>
> Overall, the results demonstrate that our method remains robust across all datasets and OOD ratios, consistently maintaining stable performance under varying degrees of semantic shift.
>
> |       Dataset       |      |      |      |      | Ratio |      |      |      |      |      |        |
> | :-----------------: | :---: | :---: | :---: | :---: | :---: | :---: | :---: | :---: | :---: | :---: | ------ |
> | **CIFAR10-C** |  0.1  |  0.2  |  0.3  |  0.4  |  0.5  |  0.6  |  0.7  |  0.8  |  0.9  |  1.0  | STD    |
> |        Noise        | 86.87 | 91.01 | 91.69 | 91.88 | 92.14 | 92.17 | 92.44 | 92.46 | 92.26 | 92.37 | 1.6947 |
> |        MNIST        | 91.13 | 92.33 | 91.39 | 92.65 | 92.59 | 92.93 | 92.38 | 91.27 | 92.47 | 91.79 | 0.6437 |
> |        SVHN        | 85.68 | 90.09 | 90.93 | 90.94 | 91.09 | 91.18 | 91.55 | 91.34 | 91.51 | 89.35 | 1.7814 |
> |        Tiny        | 76.40 | 75.49 | 73.94 | 75.14 | 75.84 | 77.18 | 78.30 | 78.89 | 79.48 | 77.92 | 1.7919 |
> |     CIFAR100-C     | 74.31 | 75.43 | 75.30 | 75.52 | 75.76 | 75.73 | 75.66 | 75.95 | 75.93 | 74.91 | 0.5075 |
>
> |       Dataset       |      |      |      |      | Ratio |      |      |      |      |      |        |
> | :------------------: | :---: | :---: | :---: | :---: | :---: | :---: | :---: | :---: | :---: | :---: | ------ |
> | **CIFAR100-C** |  0.1  |  0.2  |  0.3  |  0.4  |  0.5  |  0.6  |  0.7  |  0.8  |  0.9  |  1.0  | STD    |
> |        Noise        | 60.65 | 63.46 | 64.97 | 64.88 | 68.38 | 64.96 | 67.41 | 63.37 | 64.44 | 69.92 | 2.6900 |
> |        MNIST        | 58.37 | 61.80 | 63.26 | 60.82 | 63.99 | 64.59 | 64.19 | 63.68 | 63.46 | 62.87 | 1.8982 |
> |         SVHN         | 52.75 | 54.70 | 51.87 | 53.97 | 56.40 | 58.38 | 56.30 | 55.89 | 56.81 | 57.56 | 2.0966 |
> |         Tiny         | 47.46 | 48.93 | 48.99 | 49.65 | 50.01 | 51.16 | 51.03 | 52.03 | 52.20 | 51.89 | 1.5831 |
> |      CIFAR10-C      | 45.65 | 46.42 | 47.34 | 47.83 | 48.19 | 48.55 | 48.22 | 48.82 | 49.18 | 48.79 | 1.1283 |
>
> |       Dataset       |      |      |      |      | Ratio |      |      |      |      |      |        |
> | :------------------: | :---: | :---: | :---: | :---: | :---: | :---: | :---: | :---: | :---: | :---: | ------ |
> | **ImageNet-C** |  0.1  |  0.2  |  0.3  |  0.4  |  0.5  |  0.6  |  0.7  |  0.8  |  0.9  |  1.0  | STD    |
> |        Noise        | 62.05 | 62.99 | 62.74 | 62.93 | 62.51 | 62.74 | 62.76 | 62.74 | 62.75 | 63.06 | 0.2846 |
> |        MNIST        | 62.90 | 63.08 | 60.01 | 63.62 | 63.75 | 63.68 | 63.72 | 63.61 | 63.87 | 63.60 | 1.1564 |
> |         SVHN         | 62.28 | 62.42 | 62.45 | 62.62 | 62.46 | 62.63 | 62.60 | 62.64 | 62.76 | 62.26 | 0.1645 |
>
> |       Dataset       |      |      |      |      | Ratio |      |      |      |      |      |        |
> | :------------------: | :---: | :---: | :---: | :---: | :---: | :---: | :---: | :---: | :---: | :---: | ------ |
> | **ImageNet-R** |  0.1  |  0.2  |  0.3  |  0.4  |  0.5  |  0.6  |  0.7  |  0.8  |  0.9  |  1.0  | STD    |
> |        Noise        | 77.98 | 78.08 | 78.39 | 78.91 | 78.42 | 79.00 | 79.33 | 79.36 | 79.45 | 79.56 | 0.5894 |
> |        MNIST        | 74.78 | 75.88 | 76.00 | 76.36 | 76.14 | 76.63 | 76.59 | 76.96 | 76.95 | 77.53 | 0.7530 |
> |         SVHN         | 76.47 | 77.27 | 77.18 | 77.37 | 77.52 | 77.75 | 77.60 | 78.01 | 77.72 | 77.96 | 0.4502 |
>
> |     Dataset     |      |      |      |      | Ratio |      |      |      |      |      |        |
> | :-------------: | :---: | :---: | :---: | :---: | :---: | :---: | :---: | :---: | :---: | :---: | ------ |
> | **VisDA** |  0.1  |  0.2  |  0.3  |  0.4  |  0.5  |  0.6  |  0.7  |  0.8  |  0.9  |  1.0  | STD    |
> |      Noise      | 79.43 | 79.06 | 80.60 | 81.37 | 82.23 | 80.35 | 80.84 | 80.93 | 80.98 | 85.64 | 1.8202 |
> |      MNIST      | 79.78 | 79.66 | 79.32 | 80.31 | 80.62 | 81.10 | 80.89 | 81.53 | 81.96 | 83.16 | 1.1709 |
> |      SVHN      | 84.74 | 85.10 | 86.88 | 86.05 | 85.94 | 85.78 | 85.49 | 84.76 | 86.93 | 87.06 | 0.8725 |
>
> In the revised version, we include an additional subsection, “Analysis Under Different OOD Ratios,” in Appendix A.10 to provide a more comprehensive evaluation.

---

> > ### Author Response · Authors · 2025-11-22
> >
> > **Q2: VLBO runs both ResNet and CLIP per test sample plus clustering. Please report wall-clock and memory overhead vs. other baselines.**
> >
> > A2: Thank you for pointing this out.
> > In the revised version, we have added a dedicated section analyzing the computational overhead. Following your suggestion, we use a batch size of 128 and evaluate both the wall-clock time (s) and average memory consumption (MB) across all open-world datasets. Due to space limitations, we report here only the results using MNIST as the OOD set. The full results are provided in Section 5.4 and Appendix A.8 of the revised paper. The summarized results are shown in the table below, from which we observe that VLBO achieves a favorable balance between computational overhead and adaptation performance compared with the baselines.
> >
> > | CIFAR10-C |  Time  | Memory | $ACC_H$ | CIFAR100-C |  Time  | Memory | $ACC_H$ | ImageNet-C |  Time  | Memory | $ACC_H$ |
> > | :-------: | :----: | :-----: | :-------: | :--------: | :----: | :-----: | :-------: | :--------: | :-----: | :-----: | :-------: |
> > |   CLIP   | 0.1611 | 507.26 |   76.69   |    CLIP    | 0.1569 | 508.36 |   47.41   |    CLIP    | 0.1965 | 595.91 |   46.08   |
> > |   C-TPT   | 0.4789 | 1220.96 |   76.37   |   C-TPT   | 0.4925 | 1235.68 |   35.12   |   C-TPT   | 1.2929 | 1387.16 |   21.19   |
> > |   CLIPN   | 0.3184 | 403.25 |   32.42   |   CLIPN   | 0.3154 | 403.69 |   21.91   |   CLIPN   | 0.3054 | 408.46 |   14.11   |
> > | NegSample | 0.1130 | 578.93 |   62.53   | NegSample | 0.1147 | 579.33 |   43.34   | NegSample | 0.1246 | 598.33 |   44.05   |
> > |   WATT   | 8.8878 | 1099.38 |   76.63   |    WATT    | 7.0037 | 1099.60 |   49.82   |    WATT    | 18.3008 | 1102.08 |   47.31   |
> > |   Ours   | 0.6569 | 782.32 |   91.79   |    Ours    | 0.5425 | 912.22 |   62.87   |    Ours    | 0.9591 | 1337.59 |   63.60   |
> >
> > | ImageNet-R |  Time  | Memory | $ACC_H$ |   VisDA   |  Time  | Memory | $ACC_H$ |
> > | :--------: | :----: | :-----: | :-------: | :-------: | :----: | :-----: | :-------: |
> > |    CLIP    | 0.1877 | 593.34 |   71.05   |   CLIP   | 0.1913 | 580.69 |   74.77   |
> > |   C-TPT   | 0.5438 | 1260.39 |   68.95   |   C-TPT   | 0.4686 | 1221.33 |   75.04   |
> > |   CLIPN   | 0.3079 | 404.49 |   31.41   |   CLIPN   | 0.3041 | 403.56 |   33.37   |
> > | NegSample | 0.1195 | 596.75 |   76.39   | NegSample | 0.1143 | 596.27 |   56.63   |
> > |    WATT    | 9.0740 | 1100.16 |   70.80   |   WATT   | 8.8650 | 1099.68 |   74.86   |
> > |    Ours    | 0.9742 | 1172.48 |   77.53   |   Ours   | 0.6106 | 770.25 |   83.16   |

---

> > > ### Author Response · Authors · 2025-11-22
> > >
> > > **Q3: Missing reference on agreement-boosted between domains https://arxiv.org/abs/2406.09353. They also use confidence-based filtering for generating pseudo labels**
> > > A3: Thank you for your comment. We have added this reference to the related work section and highlighted it in blue at line 124 on Page 3. For your convenience, we reproduce the updated sentence below:
> > > “Building on this observation, several works investigate their potential within test-time adaptation settings (Osowiechi et al., 2024; Wang et al., 2024a; Phan et al., 2024).”
> > >
> > > **Q4: How does VLBO behave when OOD prevalence varies?**
> > > **A4:** Please refer to A1.

---

### Author Response · Authors · 2025-11-22
**To all Reviewers**

### To all Reviewers
We sincerely thank all reviewers for their valuable comments and constructive feedback. We have carefully addressed all the concerns and uploaded a new version of the manuscript. We summarize the major upates as below:

* **Include a diagram of our VLBO framework (Fig. 3)**. The diagram highlights the technical advances of VLBO’s two core modules: agreement-boosted filtering, which makes OOD samples more distinguishable, and semantic-boosted adaptation, which corrects false predictions by the VLM and the task-specific discriminative model through their agreement on the input image semantics. These advances have been demonstrated by empirical results (**Fig. 5-Fig. 9**)
* **Include computational overhead analysis (Section 5.4, Appendix A.8)**. It shows our VLBO achieves a more favorable balance between performance and computational overhead compared with the evaluated baselines.
* **Include analysis under different OOD ratios (Appendix A.10)**. It shows our VLBO remains consistently robust across a wide range of OOD ratios.
* **Include a table to compare our studied setting OWTTA with other related settings.** It highlights the OWTTA is the most challenging setting.

| Setting                                | Source Statistics | Domain Shift | Novel Class | Access Pattern |
| -------------------------------------- | :---------------: | :----------: | :---------: | :------------: |
| Domain Adaptation                      |        ✔️         |      ✔️      |      ✘      |   Multi-pass   |
| Source-Free Domain Adaptation          |         ✘         |      ✔️      |      ✘      |   Multi-pass   |
| Open-Set Source-Free Domain Adaptation |         ✘         |      ✔️       |     ✔️      |   Multi-pass   |
| Test-Time Domain Adaptation            |         ✘         |      ✔️      |      ✘      |  Single-pass   |
| Open-World Test-Time Adaptation        |         ✘         |      ✔️      |     ✔️      |  Single-pass   |

*Table 1: The term single-pass means that each sample can be processed only once.*

**Moreover, we highlight the novelty and key contributions of this work:** we propose VLBO , a unified framework that leverages foundation models to robustly support and enhance task-specific models in OWTTA. VLBO bridges the gap between methods that adapt foundation models alone and those that rely solely on small task-specific models. The two key modules provide a simple yet effective technical advance, as mentioned in the first point above.

---

### Comment · Area_Chair_bKtG · 2025-11-26
**Discussion Phase**

Dear reviewers,

Just a quick reminder that the discussion period will be end in a week, with reviewer replies allowed only until **Dec 2**.  At this stage, it would be very helpful if reviewers could share their thoughts on the authors’ recent clarifications, especially regarding the key points you previously highlighted. Even a short follow-up comment would help converge the discussion and ensure a fair and well-informed final decision.

Thanks!

---

### Note · Authors · 2026-01-10

I have read and agree with the venue's withdrawal policy on behalf of myself and my co-authors.